# LENSLLM: Unveiling Fine-Tuning Dynamics for LLM Selection

**Xinyue Zeng** [1]  **Haohui Wang** [1]  **Junhong Lin** [2]  **Jun Wu** [3]  **Tyler Cody** [4]  **Dawei Zhou** [1]

## Abstract

The proliferation of open-sourced Large Language Models (LLMs) and diverse downstream tasks necessitates efficient model selection, given the impracticality of fine-tuning all candidates due to computational constraints. Despite the recent advances in LLM selection, a fundamental research question largely remains nascent: *how can we model the dynamic behaviors of LLMs during fine-tuning, thereby enhancing our understanding of their generalization performance across diverse downstream tasks?* In this work, we propose a novel theoretical framework that provides a proper lens to assess the generalization capabilities of LLMs, thereby enabling accurate and efficient LLM selection for downstream applications. In particular, we first derive a *PAC-Bayesian Generalization Bound* that unveils fine-tuning dynamics of LLMs and then introduce LENSLLM, a Neural Tangent Kernel (NTK)-based Rectified Scaling Model that enables accurate performance predictions across diverse tasks while maintaining computational efficiency. Extensive empirical results on 3 large-scale benchmarks demonstrate that our model achieves up to 91.1% accuracy and reduces up to 88.5% computational cost in LLM selection, outperforming 5 state-of-the-art methods. We open-source our proposed LENSLLM model and corresponding results at LensLLM.io.

## 1. Introduction

The rise of large language models (LLMs) has revolutionized natural language processing, driving remarkable

[1]Department of Computer Science, Virginia Tech, Blacksburg, VA, USA. [2]Department of Electrical Engineering and Computer Science, Massachusetts Institute of Technology, Cambridge, MA, USA. [3]Department of Computer Science and Engineering, Michigan State University, East Lansing, MI, USA. [4]Department of Intelligent Systems Division, Virginia Tech, Blacksburg, VA, USA.. Correspondence to: Xinyue Zeng <xyzeng@vt.edu>.

*Proceedings of the 42nd International Conference on Machine Learning*, Vancouver, Canada. PMLR 267, 2025. Copyright 2025 by the author(s).

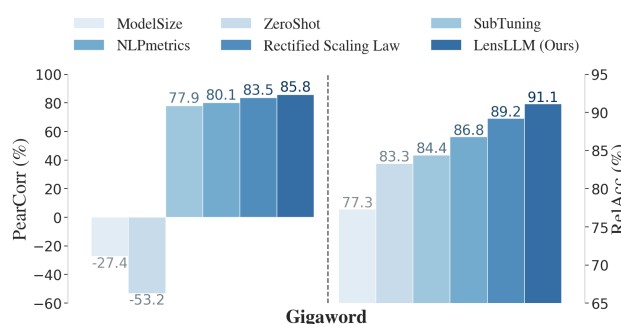

*Figure 1.* Our model demonstrates superior performance on Gigaword (See et al., 2017), achieving a Pearson Correlation Coefficient of up to 85.8% and a Relative Accuracy of up to 91.1%, surpassing 5 state-of-the-art methods for LLM selection. (Higher values indicate better performance)

progress in applications such as machine translation (Liu et al., 2020; Team et al., 2022), text summarization (Zhang et al., 2020; Lewis et al., 2019), question answering (Khashabi et al., 2020; Wei et al., 2022c), and dialogue systems (Bommasani et al., 2021; Thoppilan et al., 2022). However, the recent explosion of open-source LLMs (e.g., LLaMA (Touvron et al., 2023), Falcon (Almazrouei et al., 2023), Mistral (Mistral AI, 2024), DeepSeek (DeepSeek-AI, 2024)) has heightened interest in model selection. This selection aims to identify the optimal model from a set of candidates by balancing model complexity with the ability to explain the observed data. The tremendous variety of downstream tasks (Wei et al., 2022b; Wu et al., 2023; Wang & Duan, 2024; Jiao et al., 2023) and the exponential growth in model sizes (Zhao et al., 2024; Minaee et al., 2024), as well as the associated computational costs, have led to the need to select the optimal LLM efficiently. Moreover, the models selected through fine-tuning historical data or specific tasks may fail in novel scenarios or out-of-distribution cases (Wei et al., 2022b; Schaeffer et al., 2023), which also raises the urgent need for a generalized model selection framework for LLMs.

Model selection is a long-standing research problem in the machine-learning community. Traditional model selection approaches (Yang et al., 2023), designed for small-scale machine learning models, are inadequate for LLMs due to poor generalization and prohibitive computational costs. More recently, the machine learning community has observed in-

creasing attention on characterizing the unique behaviors of modern LLMs and developing tailored selection methods (Haowei et al., 2024). However, these methods remain mostly heuristic, often depending on strong assumptions or iterative trial-and-error processes. These limitations underscore the necessity for a theoretical grounding for LLM selection that can elucidate fine-tuning dynamics and achieve better generalization in various downstream applications.

Despite the importance of LLM selection, there are two fundamental challenges that largely remain nascent. **C1. Theoretical Understanding**: There is limited theoretical analysis characterizing the dynamic behaviors in the LLM fine-tuning process, particularly the pre-power phase that emerges in low-data regimes. A rigorous theoretical foundation is crucial for providing a proper lens in examining LLM selection and achieving model generalization. **C2. Accurate and efficient LLM Selection Framework**: There always exists a tension between model selection accuracy and computation cost. How can we find an optimal trade-off framework that could accurately select models across various tasks in resource-constrained scenarios?

In this work, we present three key contributions that advance the field of LLM selection. First, we develop a *PAC-Bayesian Generalization Bound* that reveals distinct prepower and power phases in fine-tuning. This theoretical foundation provides a novel framework for examining LLM selection, enabling robust model generalization across diverse tasks. Second, building on our theoretical results, we introduce LENSLLM, which leverages NTK (Jacot et al., 2018) to approximate the LLM fine-tuning scaling laws. By modeling the transition between pre-power and power phases, our approach not only achieves accurate LLM selection across tasks but also reduces computational costs by more than 50% compared to our best competitor Sub-Tuning (Kaplun et al., 2023). Finally, our comprehensive empirical evaluation demonstrates the superior performance of our model. For a representative example, Figure 1 shows that our model achieves up to 91.1% relative accuracy and 85.8% Pearson correlation. Moreover, it reduces computational costs by up to 88.5% compared to FullTuning while maintaining comparable performance. We open-source our proposed LENSLLM model and corresponding results at LensLLM.io.

## 2. Preliminary

In this section, we begin with the introduction of notations in Table 1, followed by scaling law behavior in LLM fine-tuning, generalization bound, and our problem definition. We consider the fine-tuning of LLMs in a supervised learning setting. The key notations used throughout this paper are summarized in Table 1. We use regular letters to denote scalars (e.g., $l$), italics letters to denote space and distribu-

*Table 1.* Notations.

| Symbol | Description |
|---|---|
| $\mathcal{X}$ | Input feature space |
| $\mathcal{D}$ | Probability distribution of samples |
| $\mathcal{M}$ | Space of LLMs |
| $n$ | Sample size |
| $l$ | Depth of neural architecture (number of layers) |
| $L$ | Expected loss on the test distribution |
| $\widehat{L}$ | Empirical loss computed on test set |
| $\boldsymbol{v}_i$ | Weight difference vector at layer $i$ |
| $\boldsymbol{x}$ | Feature vector from $\mathcal{X}$ |
| $\boldsymbol{y}$ | Targeted label of $\boldsymbol{x}$ |
| $\widehat{\boldsymbol{W}}_i$ | Pre-trained weight matrix at layer $i$ |
| $\widehat{\boldsymbol{W}}_i^{(s)}$ | Fine-tuned weight matrix at layer $i$ |
| $\boldsymbol{H}_i^+$ | Non-negative truncated Hessian matrix |

tion (e.g., $\mathcal{X}$), boldface lowercase letters to denote vectors (e.g., $\boldsymbol{v}_i$), and boldface uppercase letters to denote matrices (e.g., $\widehat{\boldsymbol{W}}_i$).

### 2.1. Scaling Laws in LLM Fine-Tuning

With the rise of LLMs, researchers in the machine-learning community have increasingly focused on understanding and characterizing the behavior of these models during the fine-tuning process. A key area of investigation is the scaling laws governing model performance, which has been extensively studied for the pre-training stage (Henighan et al., 2020; Kaplan et al., 2020; Bahri et al., 2024). These studies propose that model performance follows a predictable power-law relationship, as described below:

**Definition 1** (Power-law in Kaplan et al. (2020))**.** *The scaling loss $L(\cdot, \cdot)$ is a function of model size $N$ and training set size $D$, i.e.,*

$$L(N, D) = \left( \frac{A}{N^{\alpha_N}} + \frac{B}{D^\beta} \right)^\alpha \tag{1}$$

Here $A, B, \alpha, \alpha_N, \beta$ are universal parameters to be fitted. This groundbreaking work revealed that model performance consistently improves as a power law with increases in three critical factors: model size, training data, and computational power. These findings laid a strong foundation for understanding model behavior during pre-training.

Building on this, Haowei et al. (2024) demonstrated that fine-tuning performance depends not only on model size $N$ but also on various architectural design choices, including the number of layers, attention heads, and hidden dimensions. This intricate dependency complicates model selection using traditional scaling laws. However, to predict performance for specific models, a simplified version of the scaling law can be employed by excluding architectural considerations. Scaling laws for fixed models, as proposed in (Kaplan et al.,

2020; Hernandez et al., 2021; Tay et al., 2022), exhibit the following form:

$$L(D) = \left( \frac{B}{D^\beta} + E \right)^\alpha \tag{2}$$

where $D$ represents the training set size, while $B$, $E$, $\alpha$, and $\beta$ are parameters specific to the model and task.

As illustrated in Figure 2, two distinct phases emerge in the scaling behavior of fine-tuning test loss ($L$) with training sample size ($D$): the pre-power phase and the power phase. The pre-power phase occurs in low-data regimes where model behavior is dominated by initialization and early training dynamics. As the training size increases and reaches a transition point, models enter the power phase, characterized by predictable scaling behavior. In this phase, the relationship between training size and test loss follows a nearly linear correlation, as widely studied in prior works (Henighan et al., 2020; Kaplan et al., 2020; Bahri et al., 2024).

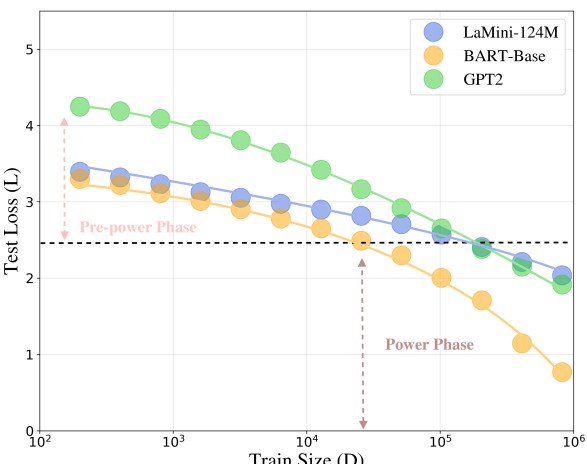

*Figure 2.* Phase transition in fine-tuning test loss ($L$) scaling with training sample size ($D$). The data reveals a pre-power phase at small $D$, followed by the established power phase showing the linear correlation between $L$ and $D$.

Understanding the underlying mechanism of this phase transition phenomenon is crucial for effective model selection. While prior work has empirically observed the emergence of power-law behavior during fine-tuning, it has largely lacked a theoretical framework to explain when and why this transition occurs. This theoretical gap limits the ability to make informed decisions about data efficiency and model scaling. In contrast, our theoretical analysis provides a first-principled understanding of the transition dynamics. First, it enables precise predictions about when additional training data will trigger the onset of the power phase, resulting in consistent and predictable performance improvements. Second, once a model enters the power phase, our scaling framework offers guidance on how to balance the costs of

further data collection against the expected performance gains.

## 2.2. Generalization Bound

While empirical studies offer valuable insights, theoretical frameworks are equally essential for understanding the dynamics of fine-tuning. Among these, generalization bound stands out as a powerful tool, providing rigorous mathematical insights into fine-tuning performance. Prior work by Ju et al. (2022) introduced a generalization bound tailored to fine-tuned feed-forward neural networks, serving as a preliminary lens for analyzing fine-tuning dynamics. However, their analysis does not explicitly account for transformer-specific architectural elements, which limits its applicability to modern large-scale language models.

**Theorem 1** (Generalization Bound in Ju et al. (2022)). *Assume the activation functions $\phi_i(\cdot)$ for all $i = 1, \ldots, l$ and the loss function $L$ are all twice-differentiable, and their first-order and second-order derivatives are all Lipschitz-continuous. Suppose $L(f_{\widehat{W}}(\boldsymbol{x}), \boldsymbol{y})$ ($L(f_{\widehat{W}})$ used in the following for simplicity) is bounded by a fixed value $C$ for any $\boldsymbol{x} \in \mathcal{X}$ with class label $\boldsymbol{y}$. Given an $l$-layer network $f_{\widehat{W}}$, with probability at least $0.99$ (or equivalently, almost everywhere in the statistical sense), for any fixed $\epsilon$ close to zero, we have*

$$L(f_{\widehat{W}}) \leq (1+\epsilon)\widehat{L}(f_{\widehat{W}}) + \frac{(1+\epsilon)\sqrt{C}\sum_{i=1}^{l}\sqrt{\mathcal{H}_i}}{\sqrt{n}} + \xi \tag{3}$$

*where $\mathcal{H}_i$ is defined as $\max_{(x,y)\in\mathcal{D}} \boldsymbol{v}_i^\top \mathbf{H}_{\mathbf{i}}^+[L(f_{\widehat{W}})]\boldsymbol{v}_i$, for all $i = 1, \ldots, l$, and $\xi = O(n^{-3/4})$ represents an error term from the Taylor's expansion.*

## 2.3. Problem Definition

In the context of LLM selection, we aim to identify the optimal model from a set of candidates for specific downstream tasks under resource-constrained scenarios. Without loss of generalization, we denote $S$ as training dataset from $\mathcal{D}$, and $\mathcal{M} = \{m_1, m_2, ..., m_k\}$ as a set of candidate models. For each model $m_i$, there is associated with a feature vector $\boldsymbol{x}_i$ representing its characteristics (e.g., model size, architecture, training data).

Given the notations above, we formally define the problem as follows:

**Problem 1.** *LLM Selection in Resource-Constrained Scenarios*
***Given:*** *(1) Limited training data $S$; (2) A set of candidate LLMs $\mathcal{M} = \{m_1, m_2, ..., m_k\}$ with their corresponding feature vectors $\boldsymbol{x}$.*
***Objective:*** *The optimal model $m^* \in \mathcal{M}$ on $S$ that has the best performance.*

# 3. Model

In this section, we introduce our framework, LENSLLM, for analyzing LLM selection. The core idea lies in modeling and regularizing the dynamics of transition phases observed during fine-tuning. Specifically, we begin by deriving a novel *PAC-Bayesian Generalization Bound* for these transition phases, incorporating the role of the Hessian matrix to capture fine-tuning behavior. Building on this theoretical foundation, we develop the overall learning paradigm of LENSLLM, focusing on effectively characterizing model behavior in both the pre-power and power phases. Finally, we present an optimization algorithm for LENSLLM, including detailed pseudo-code to illustrate its implementation.

## 3.1. Theoretical Analysis on Fine-tuning Scaling Law

Understanding the underlying mechanism of phase transitions in fine-tuning is essential for LLM selection. While empirical studies provide valuable insights, there is still a lack of theoretical foundations to reveal the dynamics of transition phases. Here we provide a theoretical foundation, *PAC-Bayesian Generalization Bound*, for understanding dynamic transition phases during LLM fine-tuning. This leads to a more nuanced analysis of fine-tuning dynamics, particularly pre-power and power phases in scaling laws. Our analysis is built upon the following key assumptions that account for transformer-specific properties:

**Assumption 1** (Smoothness). *The activation functions $\phi_i$ on layer $i$ and loss function $L$ are twice-differentiable with Lipschitz-continuous first and second derivatives (Allen-Zhu et al., 2019):*
*(1) $\|\nabla\phi_i(\boldsymbol{x}) - \nabla\phi_i(\boldsymbol{y})\| \leq L_\phi\|\boldsymbol{x} - \boldsymbol{y}\|$, where $L_\phi$ is the Lipschitz constant that bounds how fast the gradient of $\phi_i$ can change;*
*(2) $\|\nabla^2\phi_i(\boldsymbol{x}) - \nabla^2\phi_i(\boldsymbol{y})\| \leq L_{\phi'}\|\boldsymbol{x} - \boldsymbol{y}\|$, where $L_{\phi'}$ is the Lipschitz constant for the second derivative of $\phi_i$.*

**Assumption 2** (Boundedness). *Following Wei et al. (2019), the loss function and input features satisfy uniform bounds: $L \leq C$ and $\|\boldsymbol{x}\|_2 \leq B$.*

**Assumption 3** (Transformer Stability). *Based on Liu et al. (2022), transformer components satisfy:*

*(1) Attention mechanisms are $L_A$-Lipschitz continuous (Lee et al., 2020);*

*(2) Attention scores are bounded: $\|softmax(\boldsymbol{Q}\boldsymbol{K}^T)\|_\infty \leq M$ (Dong et al., 2021);*

*(3) Residual connections maintain gradient flow: $\|\nabla f(\boldsymbol{x})\| \geq \delta > 0$;*

*(4) Layer normalization preserves scale: $\|\boldsymbol{LN}(\boldsymbol{x})\|_2 = \|\boldsymbol{x}\|_2$.*

These assumptions enable us to extend Theorem 1 to trans-

formers and lead to our following main theoretical foundation:

**Theorem 2** (*PAC-Bayesian Generalization Bound*). *For a fine-tuned transformer model $f_{\hat{w}}$ satisfying Assumptions 1-3, with probability at least 0.99 and any fixed $\epsilon > 0$:*

$$L(f_{\hat{w}}) \leq (\epsilon+1)\hat{L}(f_{\hat{w}}) + \frac{(1+\epsilon)\sqrt{C}\sum_{i=1}^{l}\sqrt{h_i}}{\sqrt{n}} + O(n^{-\frac{3}{4}}) \quad (4)$$

*where $h_i \geq \max_{(x,y)\in D}\boldsymbol{v}_i^T\boldsymbol{H}_{\boldsymbol{i}}^+[L(f_{\hat{w}})]\boldsymbol{v}_i$.*

This bound provides a theoretical foundation for characterizing the fine-tuning dynamics of transformers and serves as a basis for analyzing the transition phases. The proof for this theorem is provided in Appendix A.

Understanding how language models behave during fine-tuning is essential for making informed decisions about model selection. While we have learned from empirical studies, we still lack a theoretical framework to explain the mechanism of transition between different phases in fine-tuning. This understanding could help optimize the trade-off between computational costs and model performance. To this end, we aim to use our proposed *PAC-Bayesian Generalization Bound* to explain pre-power and power phases in fine-tuning.

First, we start from the following property of the Hessian-related term $h_i$:

**Proposition 1.** *Let $\{h_i\}_{i=1}^{l}$ be a sequence of Hessian-related values that satisfy $h_i \geq \max_{(x,y)\in D}\boldsymbol{v}_i^T\boldsymbol{H}_{\boldsymbol{i}}^+[L(f_{\hat{w}})]\boldsymbol{v}_i$:*

*(1) By applying Cauchy-Schwarz theorem (Steele, 2004), we have the following inequality:*

$$\sum_{i=1}^{l}\sqrt{h_i} \leq \sqrt{l\sum_{i=1}^{l}h_i} \quad (5)$$

*(2) Based on the definition of $h_i$, we have the following upper bound:*

$$h_i \leq C_2 n^{-\beta_2} \quad (6)$$

*where $C_2$ is independent of $n$ and $\beta_2 \leq \beta_1$.*

Leveraging the above key properties, we derive the following corollary 1 to further analyze the fine-tuning dynamics. Detailed proof has been provided in Appendix B.

**Corollary 1.** *For any $\epsilon > 0$, with probability over 0.99, under Assumptions 1-3, considering the properties of the Hessian matrix, the PAC-Bayesian Generalization Bound could be extended to:*

$$L(f_{\hat{w}}) \leq (1+\epsilon)\hat{L}(f_{\hat{w}}) + C_3 n^{-\beta_3} + O(n^{-\frac{3}{4}}) \quad (7)$$

*where $C_3 = \sqrt{C \cdot l \cdot C_2}$ and $\beta_3 = \frac{\beta_2+1}{2}$ are both model/task-dependent.*

**Remark 1 (Pre-power Phase)**: The model's performance improves slowly during initial fine-tuning, with generalization error decreasing at rate $O(n^{-\frac{3}{4}})$. This phase is marked by high Hessian values and significant parameter sensitivity, necessitating careful tuning and substantial data for reliable adaptation.

**Remark 2 (Power Phase)**: As $n$ increases, the error scaling transitions to $C_3 n^{-\beta_3}$, becoming the dominant term. This phase demonstrates reduced Hessian values and enhanced stability, enabling more aggressive parameter updates and improved data efficiency.

**Remark 3 (Phase Transition)**: The transition from the pre-power phase to the power phase is marked by the change in the dominant constant factors, from $O(n^{-\frac{3}{4}})$ to $C_3 n^{-\beta_3}$, which reflects the change in the Hessian values and parameter sensitivity.

### 3.2. LENSLLM Algorithm for LLM Selection

Building upon our previous theoretical analysis of LLM fine-tuning through the lens of our *PAC-Bayesian Generalization Bound*, we now introduce LENSLLM, a novel NTK-based framework for model selection. Our approach provides a bridge between our theoretical analysis and practical model selection by leveraging the NTK to transformers so as to capture fine-tuning dynamics.

To formalize this connection, we first define an NTK matrix for a transformer model $f(\cdot, \cdot)$ with parameters $\theta$ at step $t$:

$$\boldsymbol{\Theta}_t(\boldsymbol{x}, \boldsymbol{x}') = \nabla_\theta f(\boldsymbol{x}, \theta(t)) \cdot \nabla_\theta f(\boldsymbol{x}', \theta(t)) \quad (8)$$

where $\boldsymbol{x}$ and $\boldsymbol{x}'$ represent the feature vectors of pre-training and fine-tuning, respectively. These features encapsulate relevant properties of the input samples that influence the fine-tuning process. To simplify, we use $\boldsymbol{\Theta}$ to present the NTK matrix in the following.

To further characterize the mechanism of transition phases in fine-tuning, we define the following NTK-based test loss function on transformers:

$$F(\boldsymbol{\Theta}, t) = \left|\left| e^{-\eta \boldsymbol{\Theta} t}(f_0(\boldsymbol{X}) - \boldsymbol{y}) \right|\right|_2^2 \quad (9)$$

where $\eta$ is the learning rate, $t$ represents the early stopping time in training steps during fine-tuning, $f_0(\boldsymbol{X})$ denotes initial outputs, and $\boldsymbol{y}$ represents true labels.

Motivated by these theoretical insights and their alignment with our theoretical analysis, we propose LENSLLM, a Hessian-aware rectified scaling model:

$$L(D) = \frac{B}{F(\boldsymbol{\Theta}, t) + D^\beta} + E \quad (10)$$

where $F(\Theta, t)$ is the adapted NTK-based test loss function on transformer, $D$ is the number of training data, $\beta$ denotes

the learning difficulty, B adjusts the initial test loss and E denotes the optimal loss of the model given an infinite amount of data.

The design of this loss function is intentional and serves two key purposes: (1) It models the competing effects between the transformer's intrinsic learning dynamics (captured by NTK) and the dataset size; (2) It naturally incorporates pre-trained knowledge through the initial state $f_0(\boldsymbol{X})$. In particular, the strategic placement of $F(\boldsymbol{\Theta}, t)$ in the denominator represents a significant advancement over traditional rectified scaling laws. This positioning explicitly accounts for pre-trained data influence through the NTK term, while the additive relationship between $F(\boldsymbol{\Theta}, t)$ and $D^\beta$ creates a unified framework that simultaneously captures both pre-training effects and fine-tuning data scaling.

---

**Algorithm 1** LENSLLM Algorithm

---

**Input:**
    Training subset $S$ with size $D$, models from space $\mathcal{M}$, Regression threshold $\gamma$, Stop threshold $\tau$

**Output:**
    Predicted performance score $r$ on the full dataset: $r = \exp(\psi(\log|D|))$ and minimal proportion of training data needed to reach the Pareto-Optimality curve $s = \frac{1}{2^a}$.

1: Initialize collection $\mathcal{C} = \{\}$ for (training size, test loss) pairs and $a = 1$ for iteration steps.
2: **while** TRUE **do**
3:     Train $m \in \mathcal{M}$ on $S$ and obtain NTK matrix $\boldsymbol{\Theta}$ and training steps $t$ and corresponding loss $\widehat{L}$.
4:     **if** $|\mathcal{C}| \geq \gamma$ **then**
5:         Train regression estimator $\psi$ on $\mathcal{C}$
6:         Calculate deviation set $\Delta = \{|\log \widehat{L} - \psi(\log D)|\}$ for all pairs in $\mathcal{C}$ and the standard deviation $\sigma$ of the fitting residuals
7:         **if** $s = |\log \widehat{L} - \psi(\log D)|/\sqrt{\sigma} > \tau$ **then**
8:             **break**
9:         **end if**
10:     **end if**
11:     Add $(\log|\mathcal{D}|, \log \widehat{L})$ to $\mathcal{C}$
12:     Randomly select samples from $S$ with $\frac{D}{2}$ as $S'$
13:     Update $S \leftarrow S'$
14:     Update $a \leftarrow a + 1$
15: **end while**

---

Algorithm 1 presents a two-phase performance prediction method for LLM selection through iterative dataset reduction. Starting with an empty collection $\mathcal{C}$, the algorithm trains an LLM on progressively smaller datasets and records size-loss pairs. For each iteration, it trains the model to obtain test loss $\widehat{L}$, NTK matrix $\boldsymbol{\Theta}$, and training steps $t$. After collecting $\gamma$ pairs, it fits a regression estimator $\psi$ and computes a stopping signal based on the deviation from

predicted values. The process continues until the deviation exceeds threshold $\tau$. The optimization process of our model in experiments consists of two phases: (1) a phase-fitting phase where parameter $t$ is fixed for each dataset size while optimizing $B$, $\beta$, and $E$ to fit the observed test loss, and (2) a test loss prediction phase where all parameters ($t$, $B$, $\beta$, and $E$) are jointly optimized to predict test loss across different dataset sizes and training durations. In each iteration, the dataset size is halved through random sampling until termination.

In the implementation, given a limited training dataset $S$ and a set of candidate LLMs $\mathcal{M}$, we aim to select the optimal model $m^* \in \mathcal{M}$ with the largest $r$.

## 4. Experiments

In this section, we evaluate LENSLLM's performance on LLM selection through two aspects: (a) evaluating its effectiveness against baseline models, including fitting the transition phases in fine-tuning, predicting test loss on additional training data and selecting the optimal model; (b) evaluating its efficiency by calculating computational costs. Results consistently demonstrate our model's capability in accurate and efficient LLM selection.

### 4.1. Experimental Setup

**Benchmark**: For robust evaluation across various tasks, we experiment with three benchmark datasets: FLAN (Wei et al., 2022a), Wikitext (Merity et al., 2016), and Gigaword (See et al., 2017). All of them are open-sourced on Hugging Face. To analyze performance scaling, we create smaller datasets by randomly sampling examples ranging from 200 to 1,638,400 (doubling at each step), then fine-tune and evaluate models on a separate test set.

**Baselines**: We evaluate our approach against five established baseline methods: (1) Rectified Scaling Law uses pre-trained-data-adapted scaling law model to select (Haowei et al., 2024), (2) NLP metrics uses proxy generalization metrics for selection (Yang et al., 2023), (3) SubTuning uses the performance of subset fine-tuning as selection score (Kaplun et al., 2023), (4) ModelSize uses logarithm of the number of model parameters for selection (Villalobos et al., 2022), and (5) ZeroShot uses zero-shot performance as a selection criteria (Kojima et al., 2022).

**Models**: To ensure the generality and robustness of our findings, we conduct comprehensive fine-tuning experiments across a diverse suite of pre-trained language models spanning a wide range of architectures and parameter scales. Specifically, our evaluation covers seven model families, including OPT-350M, OPT-1.3B, OPT-6.7B, T5-Small, T5-Base, Cerebras-256M, Cerebras-1.3B, mT5-Base, mT5-Large, BART-Base, BART-Large, GPT-2, LaMini-124M

and LaMini-774M. This broad selection captures variability across model sizes, training objectives, and architecture types, providing a rigorous basis for evaluating the effectiveness and stability of our fine-tuning approach.

**Optimization**: All models are fine-tuned using the AdamW optimizer with a weight decay of 0.01, and all experiments are conducted on a single NVIDIA A100 GPU with 80GB of memory. To characterize the fine-tuning dynamics across different model architectures and data sizes, we estimate $B, E, \beta, t$ for each model by minimizing the loss function:

$$\min_{B,E,\beta,t} \sum_i \left[ \text{LSE}\big( \log B - \log(F(\Theta, t) + D_i^\beta), \log E \big) - \log L(D_i) \right]$$

Where $L(D_i)$ denotes the test loss of fine-tuning on the data size $D_i$, and LSE denotes the log-exp-sum operator, used for numerical stability in modeling loss scales. This optimization allows us to capture the relationship between model capacity, data size, and fine-tuning efficacy in a unified, interpretable framework.

**Evaluation Matrics**: We use two established metrics from Haowei et al. (2024) to evaluate the performance of methods on LLM selection: Pearson correlation coefficient (PearCorr) and Relative Accuracy (RelAcc). PearCorr measures the correlation between selection scores and actual fine-tuning performance, evaluating a method's ability to rank models effectively. RelAcc is defined as the performance gap between the selected model and the best model normalized by the gap between the worst and best models.

### 4.2. Effectiveness Analysis on LLM selection

We evaluate LENSLLM's effectiveness for LLM selection through three analyses: (1) comparing its curve-fitting accuracy during fine-tuning against Rectified Scaling Law (the previous state-of-the-art method), (2) measuring its predictive performance using Root Mean Squared Error (RMSE) between predicted and actual test losses against Rectified Scaling Law, and (3) benchmarking LLM selection performance against five baseline methods (including Rectified Scaling Law) across FLAN, Wikitext, and Gigaword datasets using evaluation metrics.

**Curve Fitting**: We evaluate the curve-fitting accuracy capacity during fine-tuning. As shown in Figure 3, our model (blue square) consistently outperforms Rectified Scaling Law (red triangle) in predicting performance across architectures. For OPT-1.3b, our model provides smoother and more accurate predictions that closely track the actual test loss curve, while Rectified Scaling Law shows notable fluctuations. Our model's superiority is particularly evident in GPT-2's middle to late training stages, maintaining steady alignment with ground truth compared to Rectified Scaling Law's volatile predictions. Similarly, for T5-base, our model demonstrates more stable predictions throughout training,

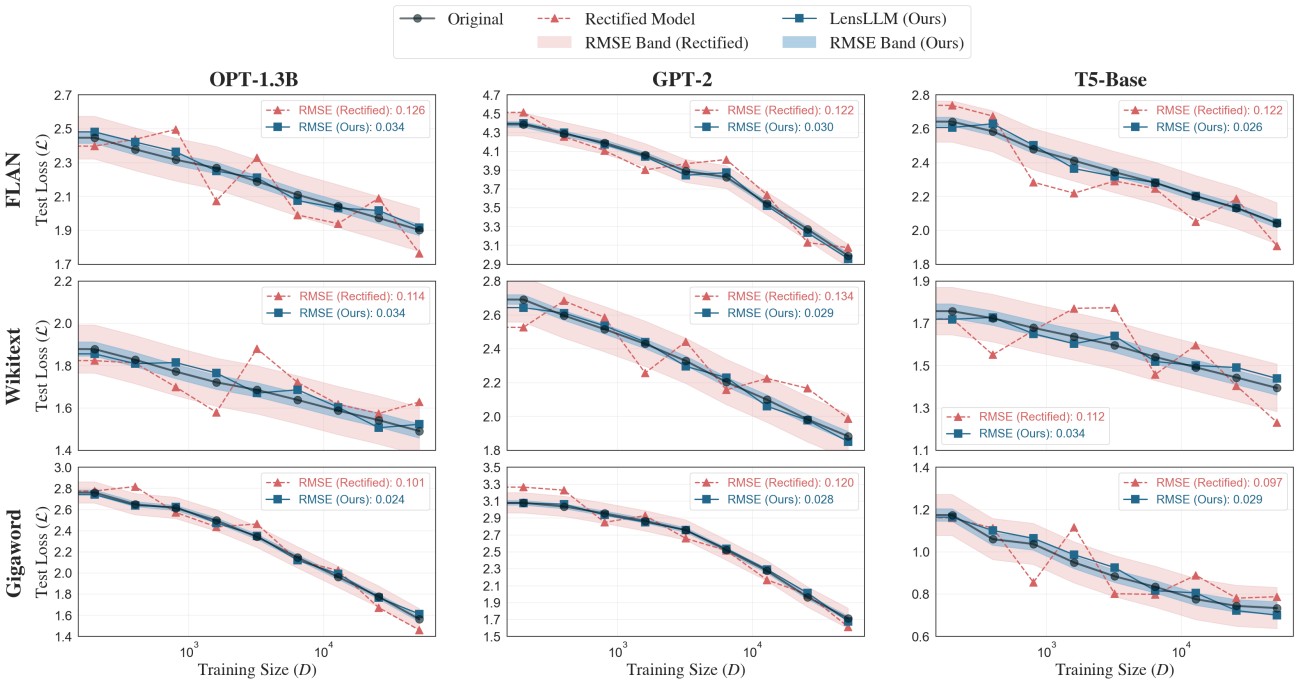

*Figure 3.* Performance comparison showing the superior effectiveness of LENSLLM (our method) across OPT-1.3b, GPT-2, and T5-base architectures on FLAN, Wikitext, and Gigaword datasets. LENSLLM consistently achieves significantly lower RMSE values (shown in a blue square) compared to the Rectified Scaling Law (shown in a red triangle), with notably smaller error bands indicating more stable performance.

effectively capturing the gradual loss descent without the oscillations seen in Rectified Scaling Law.

*Table 2.* RMSE comparison between predicted and actual test losses ($\times 10^{-1}$) of our model and Rectified Scaling Law.

| Model | Wikitext | | FLAN | | Gigaword | |
|---|---|---|---|---|---|---|
| | Ours | Rect | Ours | Rect | Ours | Rect |
| OPT-350M | **0.2** | 1.10 | **0.32** | 1.50 | **0.26** | 0.98 |
| OPT-1.3B | **0.32** | 1.14 | **0.32** | 1.20 | **0.28** | 0.99 |
| OPT-6.7B | **0.26** | 1.32 | **0.26** | 1.31 | **0.26** | 1.46 |
| T5-Small | **0.35** | 1.01 | **0.28** | 1.30 | **0.3** | 1.27 |
| T5-Base | **0.32** | 1.30 | **0.26** | 1.26 | **0.3** | 0.94 |
| Cerebras-256M | **0.24** | 1.27 | **0.22** | 1.1 | **0.33** | 1.30 |
| Cerebras-1.3B | **0.26** | 1.18 | **0.32** | 1.00 | **0.28** | 1.00 |
| mT5-Base | **0.26** | 1.17 | **0.32** | 1.22 | **0.17** | 1.07 |
| mT5-Large | **0.28** | 1.44 | **0.32** | 1.07 | **0.28** | 1.10 |
| BART-Base | **0.3** | 1.27 | **0.3** | 0.96 | **0.26** | 0.99 |
| BART-Large | **0.17** | 1.31 | **0.28** | 0.87 | **0.36** | 1.14 |
| GPT-2 | **0.3** | 1.30 | **0.3** | 1.23 | **0.26** | 1.33 |
| LaMini-124M | **0.28** | 1.01 | **0.35** | 1.00 | **0.3** | 1.15 |
| LaMini-774M | **0.32** | 1.14 | **0.28** | 1.13 | **0.28** | 1.19 |

**Test Loss Prediction**: We evaluate the test loss prediction performance of LENSLLM. Table 2 demonstrates our model's superior performance through RMSE comparisons between predicted and actual test losses. Our model

achieves significantly lower errors across all architectures: on Wikitext, errors are typically 5 times smaller (e.g., OPT-6.7B: 0.026 vs 0.132, mT5-Large: 0.028 vs 0.144); on FLAN, we maintain low RMSE (0.022-0.035) compared to Rectified Scaling Law's higher range (0.087-0.15); and on Gigaword, our model shows consistent performance below 0.036 while Rectified Scaling Law varies between 0.094-0.146. These results across three datasets and fourteen architectures confirm our model's superior accuracy in predicting training dynamics.

**LLM Selection**: We compare our approach against five baseline methods: Rectified Scaling Law, NLPmetrics, Sub-Tuning, ModelSize, and ZeroShot. All the performance is tested on a held-out validation set. Table 3 presents comprehensive model selection results across FLAN, Wikitext, and Gigaword datasets, demonstrating our method's exceptional performance. Our approach consistently achieves superior correlation (reaching 85.8% PearCorr) between predicted and actual model performance, significantly outperforming established baselines such as Rectified Scaling Law and NLPmetrics. The robust performance is further validated by outstanding RelAcc scores (up to 91.1%), indicating models selected by LENSLLM consistently approach optimal performance levels across all tasks. This substantial improvement over baseline methods establishes our frame-

*Table 3.* Model selection results (PearCorr, RelAcc) of our model (LENSLLM) and baselines (Rectified Scaling Law, NLPmetrics, SubTuning, ZeroShot and ModelSize) on three datasets in percentage. The best result within the same dataset is in **bold** font, and the second best result is underlined.

| Dataset | Metric | LENSLLM | Rectified Scaling Law | NLPmetrics | SubTuning | ZeroShot | ModelSize |
|---------|--------|---------|------------------------|------------|-----------|----------|-----------|
| FLAN | PearCorr | **78.4** | 75.2 | 76.1 | 70.5 | -12.8 | -29.8 |
|      | RelAcc | **88.3** | 85.7 | 84.7 | 80.4 | 82.3 | 75.6 |
| Wikitext | PearCorr | **82.6** | 80.3 | 78.9 | 75.4 | 6.1 | 38.0 |
|          | RelAcc | **90.2** | 88.8 | 87.5 | 85.1 | 84.4 | 68.5 |
| Gigaword | PearCorr | **85.8** | 83.5 | 80.1 | 77.9 | -53.2 | -27.4 |
|          | RelAcc | **91.1** | 89.2 | 86.8 | 84.4 | 83.3 | 77.3 |

work as a reliable guide for practitioners in optimal model selection across diverse application scenarios.

## 4.3. Efficiency Analysis on LLM Selection

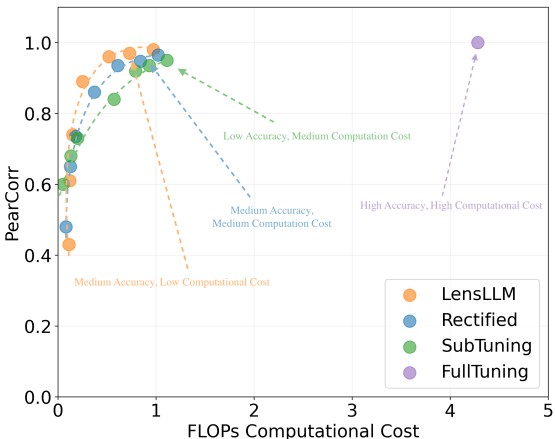

*Figure 4.* Pareto-Optimality curve between the selection performance and the computational costs (in units of $10^{21}$).(Smaller FLOPs means lower computational cost)

*Table 4.* Comparison of computational costs (in units of $10^{21}$) on benchmarks, where LENSLLM achieves superior efficiency with up to 88.5% cost reduction compared to baseline methods.

| Method | $C_{cost}$ | FLAN | Wikitext | Gigaword |
|--------|-----------|------|----------|----------|
| FullTuning | $\sum_{m \in \mathcal{M}} 6thN_m D$ | 3.35 | 5.13 | 4.28 |
| SubTuning | $\sum_{m \in \mathcal{M}} 6thN_m sD$ $= s \cdot C_{FullTuning}$ | 0.92 | 1.84 | 1.11 |
| LENSLLM | $\sum_{m \in \mathcal{M}, 2^i \leq s} 6thN_m \frac{1}{2^i} D$ $< C_{SubTuning}$ | 0.48 | 0.59 | 0.97 |

We conduct a comprehensive efficiency analysis comparing our model's computational requirements with existing methods. Following Kaplan et al. (2020), we estimate the computational cost $C_{cost}$ in floating point operations (FLOPs) as $C_{cost} \sim 6ND$, where $N$ denotes the number of model parameters and $D$ represents the dataset size. For a fair

comparison, we account for both the number of training epochs $t$ and hyper-parameter search rounds $h$ when calculating total computational costs across different tuning approaches. The performance is evaluated by the number of FLOPs when PearCorr of methods reach the curve (as illustrated in Figure 4).

As shown in Table 4, LENSLLM achieves superior computational efficiency through its innovative progressive sampling strategy: it reduces computational costs by up to 88.5% compared to FullTuning while maintaining comparable performance. LENSLLM achieves computational costs of 0.48, 0.59, and 0.97 $\times 10^{21}$ FLOPs across tasks, which substantially outperforms both SubTuning and FullTuning.

## 4.4. Ablation Study

We perform ablation studies to assess the sensitivity of our method to the stopping criteria—specifically, the regression threshold ($\gamma$) and the stop threshold ($\tau$). The following tables summarize the impact of varying these parameters on the Pearson correlation across three datasets:

*Table 5.* Impact of $\gamma$ and $\tau$ on PearCorr on FLAN

| $\gamma \backslash \tau$ | 1 | 2 | 3 | 4 | 5 |
|-------------------------|-------|-------|-------|-------|-------|
| 3 | 78.41 | 78.40 | 78.42 | 78.43 | 78.31 |
| 4 | 78.32 | 78.39 | 78.36 | 78.40 | 78.40 |
| 5 | 78.39 | 78.41 | 78.40 | 78.39 | 78.34 |

*Table 6.* Impact of $\gamma$ and $\tau$ on PearCorr on Gigaword

| $\gamma \backslash \tau$ | 1 | 2 | 3 | 4 | 5 |
|-------------------------|-------|-------|-------|-------|-------|
| 3 | 85.74 | 85.74 | 85.80 | 85.71 | 85.66 |
| 4 | 85.62 | 85.71 | 85.75 | 85.79 | 85.66 |
| 5 | 85.64 | 85.69 | 85.66 | 85.76 | 85.72 |

The results across FLAN, Gigaword, and Wikitext datasets demonstrate the robustness of our method with respect to the hyperparameters $\gamma$ and $\tau$, which govern the stopping criteria. For FLAN, the Pearson correlation remains highly stable, with minor fluctuations between 78.31 and 78.43.

*Table 7.* Impact of $\gamma$ and $\tau$ on PearCorr on Wikitext

| $\gamma \backslash \tau$ | 1 | 2 | 3 | 4 | 5 |
|---|---|---|---|---|---|
| 3 | 82.47 | 82.58 | 82.47 | 82.48 | 82.44 |
| 4 | 82.49 | 82.51 | 82.48 | 82.49 | 82.54 |
| 5 | 82.61 | 82.50 | 82.46 | 82.57 | 82.53 |

Similarly, Gigaword shows consistent performance with correlations ranging from 85.66 to 85.79. Wikitext exhibits slightly more variability, but the range (82.42 to 82.61) still reflects strong stability. These minor fluctuations across all datasets suggest that the performance of our approach is largely insensitive to moderate changes in the stopping parameters, underscoring its reliability and robustness in practical settings.

## 5. Related Work

Early model selection relied on feature similarity methods (Vu et al., 2020; Dwivedi & Roig, 2019) to predict transfer performance by comparing source and target tasks. However, these approaches couldn't capture the complex fine-tuning dynamics revealed by our NTK-based analysis, particularly the pre-power and power phases. Recent work has introduced training-free transferability metrics, such as LogME (You et al., 2021), which attempt to predict model performance without additional fine-tuning. While these approaches reduce computational overhead, they fail to account for the dynamic nature of fine-tuning that our theoretical framework explicitly models.

The development of scaling laws has been crucial in understanding LLM performance during pre-training. Kaplan et al. (2020) established foundational power-law relationships between model size, dataset size, and performance. Extensions of these scaling laws (Ghorbani et al., 2021) demonstrated their predictive power across various scales. Recent work by Isik et al. (2024) has further advanced our understanding by developing scaling laws specifically for downstream task performance, showing how different tasks exhibit distinct scaling behaviors. However, these relationships become insufficient during fine-tuning due to the emergence of distinct phases with different governing dynamics. Recent attempts to adapt scaling laws to fine-tuning, such as Rectified Scaling Law (Haowei et al., 2024), have shown promise by incorporating dataset properties and task-specific factors. Zhang et al. (2024) provides a comprehensive analysis of how data quantity, model size, and fine-tuning methods interact during LLM adaptation, revealing complex trade-offs that traditional scaling laws fail to capture. However, these approaches fall short in low-resource scenarios where the pre-power phase dominates.

Other recent methods, including learning-to-rank and transfer-learning-based approaches (Ji et al., 2024; Hu & Zhang, 2023), have focused on improving model selection efficiency through meta-learning strategies. Meanwhile, recent advances in understanding model generalization (Wang et al., 2024a;b) emphasize the importance of capturing long-tail patterns and dynamic, non-IID conditions. While these methods show promise in specific contexts, they often assume static model behaviors and fail to account for the dynamic scaling interactions.

## 6. Conclusion

This work presents a significant contribution to LLM selection by establishing a first-principled framework that bridges the gap between theoretical foundations and empirical observations. Our key contributions are threefold: First, we propose a first-principled *PAC-Bayesian Generalization Bound* that reveals the dynamics of transition phases in fine-tuning, providing a theoretical lens for examining LLM selection and its generalization across tasks. Second, building on this foundation, we introduce LENSLLM, which integrates NTK with scaling laws to better identify the transition mechanism between phases. Third, our comprehensive empirical evaluation demonstrates the exceptional performance of our approach, achieving up to $91.1\%$ relative accuracy and $85.8\%$ Pearson correlation across benchmarks—substantially outperforming existing methods. Moreover, our model achieves superior computational efficiency, which reduces computational costs by up to 88.5% compared to FullTuning. Our work establishes a new baseline for LLM selection by achieving an optimal balance between accuracy and efficiency. There might be several promising directions for future research: extending the analysis to multi-task scenarios, exploring implications for architectural design, and investigating applications to emerging model architectures, e.g. MoE models.

## Acknowledgements

We thank the anonymous reviewers for their constructive comments. This work is supported by the National Science Foundation under Award No. IIS-2339989 and No. 2406439, DARPA under contract No. HR00112490370 and No. HR001124S0013, U.S. Department of Homeland Security under Grant Award No. 17STCIN00001-08-00, Amazon-Virginia Tech Initiative for Efficient and Robust Machine Learning, Amazon AWS, Google, Cisco, 4-VA, Commonwealth Cyber Initiative, National Surface Transportation Safety Center for Excellence, and Virginia Tech. The views and conclusions are those of the authors and should not be interpreted as representing the official policies of the funding agencies or the government.

## Impact Statement

This paper advances machine learning techniques with potential applications across various domains. We acknowledge several important societal implications of this work. First, our methods could improve efficiency and accuracy in automated decision-making systems, potentially benefiting fields like healthcare and environmental monitoring. However, we also recognize potential risks, including algorithmic bias if the models are trained on non-representative data and the environmental impact of computational resources required for training. To mitigate these concerns, we provide guidelines for responsible implementation and discuss the importance of representative training data. We encourage future work to further investigate both positive and negative societal impacts, particularly regarding fairness, accountability, and environmental sustainability.

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

# A. Proof of PAC-Bayesian Generalization Bound

This section presents our proof of Theorem 2. We start with problem setup, followed by assumptions. Then we prove the theorem by following two important lemmas.

## A.1. Problem Setup

Consider predicting a target task given a training dataset of size $n$. Denote the feature vectors and labels as $\boldsymbol{x}_i$ and $y_i$, for $i = 1, \ldots, n$, in which $\boldsymbol{x}_i$ is $d$-dimensional and $y_i$ is a class label between 1 to $k$. Assume that the training examples are drawn independently from an unknown distribution $\mathcal{D}$.

We define two probability distributions:

(1)A prior distribution $\mathcal{P}$, which represents noisy perturbations around pretrained weight matrices $\mathbf{W}^{(s)}$ from the source domain.

(2)A posterior distribution $\mathcal{Q}$, which represents noisy perturbations around fine-tuned weights $\mathbf{W}$ for the target domain.

From a PAC-Bayesian perspective, the generalization performance is determined by how these noisy perturbations affect predictions across domains.

Consider an $l$-layer transformer that has been pretrained, with weight matrices $\hat{\mathbf{W}}$ for $i = 1, \ldots, l$. We then fine-tune these weights for the target task. Let $f_{\mathbf{W}}$ denote an $l$-layer transformer initialized with the pretrained weights $\hat{\mathbf{W}}^{(s)}$.

For each layer $i$, the transformer contains the following parameters:

(1) Three matrices for attention: Query projection matrix: $\mathbf{W}_i^Q$; Key projection matrix: $\mathbf{W}_i^K$; and Value projection matrix: $\mathbf{W}_i^V$.

(2) Two matrices for the feed-forward network: First layer weights: $\mathbf{W}_i^1$; and Second layer weights: $\mathbf{W}_i^2$.

(3) Layer normalization parameters: Scale parameter: $\gamma_i$; and Shift parameter: $\beta_i$.

## A.2. Assumptions

Our analysis builds upon several carefully constructed assumptions that connect classical DNN theory with transformer architectures:

**Assumption 1** (Smoothness). *The activation functions $\phi_i$ on layer $i$ and loss function $L$ are twice-differentiable with Lipschitz-continuous first and second derivatives (Allen-Zhu et al., 2019):*
*(1) $\|\nabla\phi_i(\boldsymbol{x}) - \nabla\phi_i(\boldsymbol{y})\| \leq L_\phi\|\boldsymbol{x} - \boldsymbol{y}\|$, where $L_\phi$ is the Lipschitz constant that bounds how fast the gradient of $\phi_i$ can change;*
*(2) $\|\nabla^2\phi_i(\boldsymbol{x}) - \nabla^2\phi_i(\boldsymbol{y})\| \leq L_{\phi'}\|\boldsymbol{x} - \boldsymbol{y}\|$, where $L_{\phi'}$ is the Lipschitz constant for the second derivative of $\phi_i$.*

**Assumption 2** (Boundedness). *Following Wei et al. (2019), the loss function and input features satisfy uniform bounds: $L \leq C$ and $\|\boldsymbol{x}\|_2 \leq B$.*

**Assumption 3** (Transformer Stability). *Based on Liu et al. (2022), transformer components satisfy:*

*(1) Attention mechanisms are $L_A$-Lipschitz continuous (Lee et al., 2020);*

*(2) Attention scores are bounded: $\|softmax(\boldsymbol{QK}^T)\|_\infty \leq M$ (Dong et al., 2021);*

*(3) Residual connections maintain gradient flow: $\|\nabla f(\boldsymbol{x})\| \geq \delta > 0$;*

*(4) Layer normalization preserves scale: $\|\boldsymbol{LN}(\boldsymbol{x})\|_2 = \|\boldsymbol{x}\|_2$.*

## A.3. Proofs for PAC-Bayesian Generalization Bound

Before we start to prove Theorem 2, we will need the KL divergence between the prior P and the posterior Q in the PAC-Bayesian analysis. This is stated in the following result:

**Proposition 2** (KL Divergence for Transformer). *Suppose the noise perturbation at layer $i$ is drawn from a Gaussian distribution with mean zero and covariance $\Sigma_i$, for every $i = 1, \ldots, l$. Then:*

*1. The KL divergence between $\mathcal{P}$ and $\mathcal{Q}$ is:*

$$KL(\mathcal{Q}\|\mathcal{P}) = \frac{1}{2}\sum_{i=1}^{l}(\boldsymbol{W}_i - \boldsymbol{W}_i^{(s)})^\top \boldsymbol{\Sigma}_i^{-1}(\boldsymbol{W}_i - \boldsymbol{W}_i^{(s)}) \tag{11}$$

*2. For isotropic noise distribution at every layer (i.e., $\boldsymbol{\Sigma}_i = \sigma_i^2 \boldsymbol{I}_d$):*

$$KL(\mathcal{Q}\|\mathcal{P}) = \sum_{i=1}^{l} \frac{\|\boldsymbol{W}_i - \boldsymbol{W}_i^{(s)}\|_F^2}{2\sigma_i^2} \tag{12}$$

*Proof.* The proof follows from standard results on multivariate normal distributions with additional attention to transformer components.

Let $\boldsymbol{Z}_i$ be the weight matrix of layer $i$ in the posterior distribution. By definition of KL divergence:

$$
\begin{aligned}
KL(\mathcal{Q}\|\mathcal{P}) &= \mathbb{E}_{\boldsymbol{Z}\sim\mathcal{Q}}\left[\log\frac{\mathcal{Q}(\boldsymbol{Z})}{\mathcal{P}(\boldsymbol{Z})}\right] \\
&= \mathbb{E}_{\boldsymbol{Z}\sim\mathcal{Q}}[\log\mathcal{Q}(\boldsymbol{Z}) - \log\mathcal{P}(\boldsymbol{Z})] \\
&= \mathbb{E}_{\boldsymbol{Z}\sim\mathcal{Q}}\left[\sum_{i=1}^{l}\left(-\frac{1}{2}vec(\boldsymbol{Z}_i - \boldsymbol{W}_i)^\top\boldsymbol{\Sigma}_i^{-1}vec(\boldsymbol{Z}_i - \boldsymbol{W}_i) + \frac{1}{2}vec(\boldsymbol{Z}_i - \hat{\boldsymbol{W}}_i^{(s)})^\top\boldsymbol{\Sigma}_i^{-1}vec(\boldsymbol{Z}_i - \hat{\boldsymbol{W}}_i^{(s)})\right)\right]
\end{aligned}
$$

wheer $\boldsymbol{Z}_i$ includes both feed-forward and attention parameters on layer $i$. Computing the expectation over $\boldsymbol{Z}i$ and using matrix trace properties:

$$KL(\mathcal{Q}\|\mathcal{P}) = -\frac{1}{2}\mathbb{E}_{\boldsymbol{Z}\sim\mathcal{Q}}\left[\sum_{i=1}^{l}\mathrm{Tr}\left[vec(\boldsymbol{Z}_i - \boldsymbol{W}_i)vec(\boldsymbol{Z}_i - \boldsymbol{W}i)^\top\Sigma i^{-1}\right] - \mathrm{Tr}\left[vec(\boldsymbol{Z}_i - \hat{\boldsymbol{W}}_i^{(s)})vec(\boldsymbol{Z}_i - \hat{\boldsymbol{W}}_i^{(s)})^\top\boldsymbol{\Sigma}_i^{-1}\right]\right]$$

Given that the expectation of $\boldsymbol{Z}_i$ is $\boldsymbol{W}_i$ and covariance is $\Sigma i$, after cancellation:

$$KL(\mathcal{Q}\|\mathcal{P}) = \frac{1}{2}\sum_{i=1}^{l}vec(\boldsymbol{W}_i - \hat{\boldsymbol{W}}_i^{(s)})^\top\boldsymbol{\Sigma}_i^{-1}vec(\boldsymbol{W}_i - \hat{\boldsymbol{W}}_i^{(s)})$$

For isotropic case where $\boldsymbol{\Sigma}_i = \sigma_i^2\mathcal{I}d$, this simplifies to:

$$KL(\mathcal{Q}\|\mathcal{P}) = \sum_{i=1}^{l}\frac{\|\boldsymbol{W}_i - \boldsymbol{W}_i^{(s)}\|_F^2}{2\sigma_i^2}$$

$\square$

**Theorem 2** (*PAC-Bayesian Generalization Bound*). *For a fine-tuned transformer model $f_{\hat{w}}$ satisfying Assumptions 1-3, with probability at least 0.99 and any fixed $\epsilon > 0$:*

$$L(f_{\hat{w}}) \leq (\epsilon+1)\hat{L}(f_{\hat{w}}) + \frac{(1+\epsilon)\sqrt{C}\sum_{i=1}^{l}\sqrt{h_i}}{\sqrt{n}} + O(n^{-\frac{3}{4}}) \tag{4}$$

*where $h_i \geq \max_{(x,y)\in D}\boldsymbol{v}_i^T\boldsymbol{H}_{\boldsymbol{i}}^{+}[L(f_{\hat{w}})]\boldsymbol{v}_i$.*

*Proof.* First, we separate the gap between $L(f_{\hat{w}})$ and $\frac{1}{\beta}\hat{L}(f_{\hat{w}})$ into three parts:

$$L(f_{\hat{w}}) - \frac{1}{\beta}\hat{L}(f_{\hat{w}}) = L(f_{\hat{w}}) - L_{\mathcal{Q}}(f_{\hat{w}}) + L_{\mathcal{Q}}(f_{\hat{w}}) - \frac{1}{\beta}\hat{L}_{\mathcal{Q}}(f_{\hat{w}}) + \frac{1}{\beta}\hat{L}_{\mathcal{Q}}(f_{\hat{w}}) - \frac{1}{\beta}\hat{L}(f_{\hat{w}}). \tag{13}$$

By Taylor's expansion, we can bound Equation 13 with respect to the empirical loss and the expected loss:

$$L(f_{\hat{w}}) - \frac{1}{\beta}\hat{L}(f_{\hat{w}}) \leq -\mathbb{E}_{(x,y)\sim\mathcal{D}}\left[\sum_{i=1}^{l}\langle\nabla_i\boldsymbol{H}_i\left[\ell(f_{\hat{w}}(x),y)\right]\rangle\right] + \sum_{i=1}^{l}C_i\left\|\nabla_i\boldsymbol{H}_i\right\|_F^{\frac{3}{2}} + \left(L_{\mathcal{Q}}(f_{\hat{w}}) - \frac{1}{\beta}\hat{L}_{\mathcal{Q}}(f_{\hat{w}})\right)$$

$$+ \frac{1}{\beta}\left(\frac{1}{n}\sum_{i=1}^{l}\sum_{j=1}^{n}\langle\nabla_i\boldsymbol{H}_i\left[\ell(f_{\hat{w}}(x_j),y_j)\right]\rangle\right) + \sum_{i=1}^{l}C_i\left\|\nabla_i\boldsymbol{H}_i\right\|_F^{\frac{3}{2}},$$

which simplifies to:

$$L(f_{\hat{w}}) - \frac{1}{\beta}\hat{L}(f_{\hat{w}}) \leq -\mathbb{E}_{(x,y)\sim\mathcal{D}}\left[\sum_{i=1}^{l}\langle\nabla_i\boldsymbol{H}_i\left[\ell(f_{\hat{w}}(x),y)\right]\rangle\right] + \frac{1}{n\beta}\sum_{i=1}^{l}\sum_{j=1}^{n}\langle\nabla_i\boldsymbol{H}_i\left[\ell(f_{\hat{w}}(x_j),y_j)\right]\rangle$$

$$+ \left(\frac{1}{\beta}+1\right)\sum_{i=1}^{l}C_i\left\|\nabla_i\boldsymbol{H}_i\right\|_F^{\frac{3}{2}} + \left(L_{\mathcal{Q}}(f_{\hat{w}}) - \frac{1}{\beta}\hat{L}_{\mathcal{Q}}(f_{\hat{w}})\right).$$

Next, we combine the upper bound on the noise stability of $f_{\hat{w}}$ with respect to the empirical loss and the expected loss:

$$\frac{1}{n\beta}\sum_{i=1}^{l}\sum_{j=1}^{n}\langle\nabla_i\boldsymbol{H}_i\left[\ell(f_{\hat{w}}(x_j),y_j)\right]\rangle - \mathbb{E}_{(x,y)\sim\mathcal{D}}\left[\sum_{i=1}^{l}\langle\nabla_i\boldsymbol{H}_i\left[\ell(f_{\hat{w}}(x),y)\right]\rangle\right]$$

$$= \frac{1}{\beta}\sum_{i=1}^{l}\left(\frac{1}{n}\sum_{j=1}^{n}\langle\nabla_i\boldsymbol{H}_i\left[\ell(f_{\hat{w}}(x_j),y_j)\right]\rangle - \mathbb{E}_{(x,y)\sim\mathcal{D}}\left[\langle\nabla_i\boldsymbol{H}_i\left[\ell(f_{\hat{w}}(x),y)\right]\rangle\right]\right)$$

$$+ \left(\frac{1}{\beta}-1\right)\sum_{i=1}^{l}\nabla_i\langle\mathbb{E}_{(x,y)\sim\mathcal{D}}\langle\boldsymbol{H}_i\left[\ell(f_{\hat{w}}(x),y)\rangle\right]\rangle.$$

Recall that $\boldsymbol{v}_i$ is a flattened vector of the matrix $\hat{\boldsymbol{W}}_i - \hat{\boldsymbol{W}}_i^{(s)}$. By the assumptions 1-3 and the KL divergence proposition,

$$L(f_{\hat{w}}) - \frac{1}{\beta}\hat{L}(f_{\hat{w}}) \leq \frac{C(KL(\mathcal{Q}\|\mathcal{P}) + \log\frac{1}{\delta})}{2\beta(1-\beta)n}$$

$$\leq \frac{C\left(\frac{1}{2}\sum_{i=1}^{l}\langle\boldsymbol{v}_i,\boldsymbol{\Sigma}_i^{-1}\boldsymbol{v}_i\rangle + \log\frac{1}{\delta}\right)}{2\beta(1-\beta)n}.$$

Combining all above equations with probability at least $1 - 2\delta$, we get the following Inequation 14:

$$L(f_{\hat{w}}) - \frac{1}{\beta}\hat{L}(f_{\hat{w}}) \leq \frac{C_2\sqrt{\log(C_3 n/\delta)/n}}{\beta}\sum_{i=1}^{l}\left\|\nabla_i\boldsymbol{H}_i\right\|_F + \left(\frac{1}{\beta}-1\right)\sum_{i=1}^{l}\nabla_i\langle\mathbb{E}_{(x,y)\sim\mathcal{D}}\langle\boldsymbol{H}_i\left[\ell(f_{\hat{w}}(x),y)\rangle\right]\rangle$$

$$+ \left(\frac{1}{\beta}+1\right)C_1\sum_{i=1}^{l}\left\|\nabla_i\boldsymbol{H}_i\right\|_F^{\frac{3}{2}} + \frac{C\left(\frac{1}{2}\sum_{i=1}^{l}\langle\boldsymbol{v}_i,\boldsymbol{\Sigma}_i^{-1}\boldsymbol{v}_i\rangle + \log\frac{1}{\delta}\right)}{2\beta(1-\beta)n}. \tag{14}$$

Recall the truncated Hessian $\boldsymbol{H}_i^+\left[\ell(f_{\hat{w}}(\boldsymbol{x}), \boldsymbol{y})\right]$ is equal to $\boldsymbol{U}_i \max(\boldsymbol{D}_i, 0)\boldsymbol{U}_i^\top$, where $\boldsymbol{U}_i \boldsymbol{D}_i \boldsymbol{U}_i^\top$ is the eigen-decomposition of $\boldsymbol{H}_i\left[\ell(f_{\hat{w}}(\boldsymbol{x}), \boldsymbol{y})\right]$. For any $(\boldsymbol{x}, \boldsymbol{y}) \sim \mathcal{D}$, we have:

$$\nabla_i \boldsymbol{H}_i\left[\ell(f_{\hat{w}}(\boldsymbol{x}), \boldsymbol{y})\right] \leq \boldsymbol{H}_i^+\left[\ell(f_{\hat{w}}(\boldsymbol{x}), \boldsymbol{y})\right]$$

.

Thus, after taking $\mathbb{E}_{(\boldsymbol{x}, \boldsymbol{y}) \sim \mathcal{D}}$ on both sides, we have the following inequation:

$$\left(\frac{1}{\beta} - 1\right) \sum_{i=1}^l \nabla_i \langle \mathbb{E}_{(x,y) \sim \mathcal{D}} \langle \boldsymbol{H}_i\left[\ell(f_{\hat{w}}(x), y))\right] \rangle \rangle = \sqrt{\frac{C}{4\beta^2 n \|\boldsymbol{v}_i\|^2}} \langle \mathbb{E}_{(\boldsymbol{x}, \boldsymbol{y}) \sim \mathcal{D}} \left[\boldsymbol{H}_i^+\left[\ell(f_{\hat{w}}(\boldsymbol{x}), \boldsymbol{y})\right]\right]^{\frac{1}{2}}, \boldsymbol{v}_i \boldsymbol{v}_i^T \rangle$$

$$\leq \sqrt{\frac{C}{4\beta^2 n \|\boldsymbol{v}_i\|^2}} \|\mathbb{E}_{(\boldsymbol{x}, \boldsymbol{y}) \sim \mathcal{D}} \left[\boldsymbol{H}_i^+\left[\ell(f_{\hat{w}}(\boldsymbol{x}), \boldsymbol{y})\right]\right]^{\frac{1}{2}} \boldsymbol{v}_i\| \|\dot{\boldsymbol{v}}_i^T\| \quad (15)$$

$$= \sqrt{\frac{C\dot{\boldsymbol{v}}_i^T \mathbb{E}_{(\boldsymbol{x}, \boldsymbol{y}) \sim \mathcal{D}} \left[\boldsymbol{H}_i^+\left[\ell(f_{\hat{w}}(\boldsymbol{x}), \boldsymbol{y})\right]\right] \boldsymbol{v}_i}{4\beta^2 n}} \quad \leq \sum_{i=1}^l \sqrt{\frac{C\mathcal{H}_i}{\beta^2 n}}$$

Next, based on the Inequation 15, the upper bound of $L(f_{\hat{w}}) - \frac{1}{\beta}\hat{L}(f_{\hat{w}})$ could be extended to the following:

$$L(f_{\hat{w}}) - \frac{1}{\beta}\hat{L}(f_{\hat{w}})$$

$$\leq \sum_{i=1}^l \sqrt{\frac{C\mathcal{H}_i}{\beta^2 n}} + \left(C_2 \sqrt{\frac{\log(C_3 n/\delta)/n}{\beta}} \sum_{i=1}^l \|\nabla_i \boldsymbol{H}_i\|_F + \left(1 + \frac{1}{\beta}\right) C_1 \sum_{i=1}^l \|\nabla_i \boldsymbol{H}_i\|_F^{\frac{3}{2}} + \frac{C}{2\beta(1-\beta)n} \log\frac{1}{\delta}\right)$$

Then we set $\epsilon = (1 - \beta)/\beta$, and get this:

$$L(f_{\hat{w}}) \leq (1 + \epsilon)(\hat{L}(f_{\hat{w}}) + \sum_{i=1}^l \sqrt{\frac{C\mathcal{H}_i}{\beta^2 n}})$$

$$+ \left(C_2 \sqrt{\frac{\log(C_3 n/\delta)/n}{\beta}} \sum_{i=1}^l \|\nabla_i \boldsymbol{H}_i\|_F + \left(1 + \frac{1}{\beta}\right) C_1 \sum_{i=1}^l \|\nabla_i \boldsymbol{H}_i\|_F^{3/2} + \frac{C}{2\beta(1-\beta)n} \log\frac{1}{\delta}\right).$$

where the last part is equal to $O(n^{-\frac{3}{4}})$.

Hence, we can conclude that:

$$L(f_{\hat{w}}) \leq (1 + \epsilon)(\hat{L}(f_{\hat{w}}) + \sum_{i=1}^l \sqrt{\frac{C\mathcal{H}_i}{\beta^2 n}}) + O(n^{-\frac{3}{4}}). \quad (16)$$

Thus we have shown the Theorem 2 holds.

$$\square$$

This result shows that transformer architectures maintain PAC-Bayesian generalization guarantees with additional terms accounting for attention mechanisms, layer normalization, and residual connections.

## B. Proof of Scaling Behavior in PAC-Bayesian Generalization Bound

In this appendix, we provide the detailed proofs for corollary 1.

**B.1. Property 1: Cauchy-Schwarz on $h_i$**

**Lemma 1.** $\sum_{i=1}^{n} \sqrt{h_i} \leq \sqrt{n \sum_{i=1}^{n} h_i}$

*Proof.* The proof follows these key steps:

1) First, recall the Cauchy-Schwarz inequality: For vectors $\boldsymbol{u}, \boldsymbol{v} \in \mathbb{R}^n$, $|\langle \boldsymbol{u}, \boldsymbol{v} \rangle| \leq \|\boldsymbol{u}\|\|\boldsymbol{v}\|$.

2) Let's define our vectors:

- $\boldsymbol{u} = (\sqrt{h_1}, \sqrt{h_2}, \ldots, \sqrt{h_n})$

- $\boldsymbol{v} = (1, 1, \ldots, 1)$ ($n$-dimensional vector of ones)

3) Calculate the left side $|\langle \boldsymbol{u}, \boldsymbol{v} \rangle|$:

$$\langle \boldsymbol{u}, \boldsymbol{v} \rangle = \sum_{i=1}^{n} (\sqrt{h_i} \cdot 1)$$
$$= \sum_{i=1}^{n} \sqrt{h_i}$$

4) Calculate $\|\boldsymbol{u}\|$:

$$\|\boldsymbol{u}\| = \sqrt{\sum_{i=1}^{n} (\sqrt{h_i})^2}$$
$$= \sqrt{\sum_{i=1}^{n} h_i}$$

5) Calculate $\|\boldsymbol{v}\|$:

$$\|\boldsymbol{v}\| = \sqrt{\sum_{i=1}^{n} 1^2}$$
$$= \sqrt{n}$$

6) Apply Cauchy-Schwarz:

$$|\langle \boldsymbol{u}, \boldsymbol{v} \rangle| \leq \|\boldsymbol{u}\|\|\boldsymbol{v}\|$$
$$\sum_{i=1}^{n} \sqrt{h_i} \leq \sqrt{\sum_{i=1}^{n} h_i} \cdot \sqrt{n}$$
$$\sum_{i=1}^{n} \sqrt{h_i} \leq \sqrt{n \sum_{i=1}^{n} h_i}$$

Therefore, we have proven that $\sum_{i=1}^{n} \sqrt{h_i} \leq \sqrt{n \sum_{i=1}^{n} h_i}$. $\qquad\square$

**Note:** This proof assumes all $h_i$ are non-negative real numbers, which is aligned with the property of $h_i$ in our bound.

**B.2. Property 2: Upper Bound of $h_i$**

We first prove that the sum of the trace of the Hessian matrix of each layer $\mathrm{tr}(\boldsymbol{H}_l)$ in a transformer model, when summed across all layers, is proportional to $n^{-\beta}$, where $n$ is the size of the fine-tuning dataset and $\beta$ is a constant.

B.2.1. SUM OF TRACE OF $\boldsymbol{H_i}$

**1. Define the Objective Function and Hessian**

- Let $L(\theta)$ be the loss function of the model parameterized by $\theta$.

- The Hessian matrix for this loss function is defined as:

$$\boldsymbol{H} = \nabla_\theta^2 L(\theta) \tag{17}$$

  For each layer $l$, let $\boldsymbol{H}_l$ be the Hessian of the loss function with respect to the parameters in that layer.

- The sum of the trace of the Hessian matrix across all layers is:

$$\sum_{l=1}^{L} \text{tr}(\boldsymbol{H}_l) = \text{tr}(\boldsymbol{H}) \tag{18}$$

**2. Scaling Behavior of the Hessian with Respect to Dataset Size**

**Lemma 2.** *If $L(\theta)$ is the empirical loss over a dataset of size $n$, the trace of the Hessian matrix $\boldsymbol{H}$ scales as $tr(\boldsymbol{H}) = n^{-1} Var(\nabla L(\theta))$.*

*Proof.* • Consider the empirical loss function:

$$L(\theta) = \frac{1}{n} \sum_{i=1}^{n} \ell(\theta; x_i) \tag{19}$$

  where $\ell(\theta; x_i)$ is the loss associated with sample $x_i$.

- The Hessian of this loss function can be expressed as:

$$\boldsymbol{H} = \frac{1}{n} \sum_{i=1}^{n} \boldsymbol{H}_i \tag{20}$$

  where $\boldsymbol{H}_i = \nabla_\theta^2 \ell(\theta; x_i)$ is the Hessian of the individual sample loss.

- By taking the trace, we have:

$$\text{tr}(\boldsymbol{H}) = \frac{1}{n} \sum_{i=1}^{n} \text{tr}(\boldsymbol{H}_i) \tag{21}$$

- Using the Central Limit Theorem (CLT) and assuming that $\boldsymbol{H}_i$ are i.i.d., the variance of the gradient $\nabla \ell(\theta; x_i)$ becomes:

$$\text{Var}(\nabla L(\theta)) = \frac{1}{n} \text{Var}(\nabla \ell(\theta)) \tag{22}$$

- From Dauphin et al. (2024), we have the following relation between the trace of $\boldsymbol{H}$ and $\text{Var}(\nabla L(\theta))$:

$$\text{tr}(\boldsymbol{H}) = \text{Var}(\nabla L(\theta)) + \text{tr}\left(\nabla_z L \cdot \nabla_\theta^2 \mathbf{z}\right) \tag{23}$$

  if we assume that the model "fits" the data, w.r.t $\nabla_\theta L(\theta^*) = 0$, then we could get the following result:

$$\text{tr}(\boldsymbol{H}) = \text{Var}(\nabla L(\theta)) \tag{24}$$

  That is:

$$\text{tr}(\boldsymbol{H}) = n^{-1} \text{Var}(\nabla L(\theta)) \tag{25}$$

$\square$

## 3. The behavior of Variance During Fine-Tuning

**Lemma 3.** *During fine-tuning, the variance of the gradient scales as $Var(\nabla L(\theta)) \propto n^{\alpha}$ for some constant $\alpha$.*

*Proof.* • Let the variance of the gradient during fine-tuning be $\sigma^2(n)$.

- Empirical observations and theoretical results from the literature suggest that as the size of the dataset increases, the variance of the gradient decreases. By analyzing the dynamics of SGD through a Bayesian lens, this behavior is often modeled as Smith & Le (2018):

$$\sigma^2(n) \propto n^{-\alpha} \tag{26}$$

The parameter $\alpha$ characterizes the behavior of the variance reduction. It is typically derived based on the structure of the model and the type of regularization applied.

$\square$

## 4. Combining Lemmas 1 and 2

- From Lemma 1:

$$\text{tr}(\boldsymbol{H}) = n^{-1}\sigma^2(n) \tag{27}$$

- Substituting the result from Lemma 2:

$$\text{tr}(\boldsymbol{H}) \propto n^{-1} \cdot n^{-\alpha} = n^{-\alpha-1} \tag{28}$$

The sum of the trace of the Hessian matrix across all layers is proportional to $n^{-\beta}$, where $\beta = \alpha + 1$. Let's give $\text{tr}(\boldsymbol{H}) = C_1 n^{-\beta_1}$ as the conclusion of this statement.

### B.2.2. UPPER BOUND OF $h_i$

Based on our proof in Appendix A, we have the following formula for a L-layer transformer:

$$L(f_{\hat{w}}) \leq (\epsilon + 1)\hat{L}(f_{\hat{w}}) + \frac{(1+\epsilon)\sqrt{C}\sum_{i=1}^{L}\sqrt{h_i}}{\sqrt{n}} + \xi \tag{29}$$

where $h_i$ is any value greater than $\max_{(x,y)\in D} v_i^T \boldsymbol{H}_i^+ [\ell(f_{\hat{w}}(x), y)] v_i$, for all $i = 1, \ldots, L$.

To make the proof more straightforward and clear, we would start our proof on a 2-layer transformer as follows:

$$L(f_{\hat{w}}) \leq (\epsilon + 1)\hat{L}(f_{\hat{w}}) + \frac{(1+\epsilon)\sqrt{C}\sum_{i=1}^{2}\sqrt{h_i}}{\sqrt{n}} + \xi \tag{30}$$

where $h_i$ is any value greater than $\max_{(x,y)\in D} v_i^T \boldsymbol{H}_i^+ [\ell(f_{\hat{w}}(x), y)] v_i$, for all $i = 1, \ldots, 2$.

We then prove that the sum of $h_i$ has an upper bound $C_2 n^{-\beta_2}$, given the distance-based generalization, where $C_2$ is independent of $n$.

### 1. Define the Objective Function and Weight Matrix

- Let $\hat{\boldsymbol{W}}^{(s)}$ be the weight matrices of pre-trained model and $\boldsymbol{W}_i$ be the dimension of layer $i$. The dimension of $\boldsymbol{W}_i$ is $d_i$ by $d_{i-1}$, where $d_i$ is the dimension of input $x_i$.

- The distance-based regularization is defined as for every layer:

$$\|\boldsymbol{W}_i - \hat{\boldsymbol{W}}_i^{(s)}\|_F \leq \alpha_i, \quad \forall i = 1, 2. \tag{31}$$

- Therefore, $h_i$ can be forward defined as:

$$h_i \leq \|\boldsymbol{W}_i - \hat{\boldsymbol{W}}_i^{(s)}\|_F^2 \max_{(x,y)\in\mathcal{X}} \text{Tr}[H_i(\ell(f_{\hat{w}}(x), y))] \tag{32}$$

**2. Upper Bound for $\|\boldsymbol{W}_i - \hat{\boldsymbol{W}}_i^{(s)}\|_F^2$**

**Lemma 4.** *There exists an upper bound for $\|\boldsymbol{W}_i - \hat{\boldsymbol{W}}_i^{(s)}\|_F^2$, which is unrelated to the training data size:*

$$\|\boldsymbol{W}_i - \hat{\boldsymbol{W}}_i^{(s)}\|_F^2 \leq B \tag{33}$$

*Proof.* • Trauger & Tewari (2023) has proved that for any $t \in \mathbb{N}$:

$$\|(\boldsymbol{W} - \hat{\boldsymbol{W}})\boldsymbol{x}_t\|_q^q = \sum_{j=1}^{k}((\boldsymbol{W} - \hat{\boldsymbol{W}}\boldsymbol{x}_t)^q \leq k\epsilon^q \tag{34}$$

• From which, we could derive:

$$\|(\boldsymbol{W} - \hat{\boldsymbol{W}})\boldsymbol{x}_t\|_2^2 \leq k\epsilon^2 \tag{35}$$

• For any $\boldsymbol{x}_i \neq \emptyset$ we have:

$$\|(\boldsymbol{W} - \hat{\boldsymbol{W}})\|_2^2 \leq \frac{k\epsilon^2}{min(\boldsymbol{x}_i\boldsymbol{x}_i^T)} \tag{36}$$

in our setting.

• Thus, considering the relation between Frobenius Norm and Spectral Norm, we have:

$$\|(\boldsymbol{W} - \hat{\boldsymbol{W}}_i^{(s)})\|_2^2 \leq \frac{\sigma^2}{min(\boldsymbol{x}_i\boldsymbol{x}_i^T)} \tag{37}$$

• Then, we have:

$$\|\boldsymbol{W}_i - \hat{\boldsymbol{W}}_i^{(s)}\|_F^2 \leq min\{d_i, d_{i-1}\}\frac{\sigma^2}{\boldsymbol{x}_i\boldsymbol{x}_i^T} = B \tag{38}$$

Since none of $d_i$, $\sigma$ and $\boldsymbol{x}_i$ relies on training data size n, $C_2$ is independent of n.

• When $\boldsymbol{x}_i = \emptyset$, we have:

$$\|(\boldsymbol{W}_i - \hat{\boldsymbol{W}}_i^s)\|_2^2 = 0 \tag{39}$$

This is because there is no training data for fine-tuning, the weights would not change, and 0 is independent of $n$.

• Thus,

$$\|(\boldsymbol{W}_i - \hat{\boldsymbol{W}}_i^s)\|_2^2 \leq B \tag{40}$$

works for any $x_i$.

$\square$

Since $\|\boldsymbol{W}_i - \hat{\boldsymbol{W}}_i^{(s)}\|_F^2$ has upper bound of $B$, we can derive a upper bound of $h_i$ as $B \max_{(x,y)\in\mathcal{X}} \text{Tr}[\boldsymbol{H}_i(\ell(f_{\hat{w}}(x), y))]$. Using the result of statement 1, we can continually derive the upper bound as:

$$h_i \leq C_2 n^{-\beta_2} \tag{41}$$

### B.3. Fine-tuning Scaling Law

Since $\boldsymbol{H}_i$ is always larger than 0, then we can extend the above bound to the following one by using the relation of $\sum_{i=1}^{2} \sqrt{h_i} \leq \sqrt{2\sum_{i=1}^{2} h_i}$:

$$L(f_{\hat{w}}) \leq (\epsilon + 1)\hat{L}(f_{\hat{w}}) + \frac{(1 + \epsilon)\sqrt{C}\sqrt{2\sum_{i=1}^{2} h_i}}{\sqrt{n}} + \xi \tag{42}$$

From the above statements, we finally reach our conclusion as follows:

$$
\begin{aligned}
L(f_{\hat{w}}) &\leq (\epsilon+1)\hat{L}(f_{\hat{w}}) + \frac{(1+\epsilon)\sqrt{C}\sqrt{2\sum_{i=1}^{2}h_i}}{\sqrt{n}} + \xi \\
&\leq (\epsilon+1)\hat{L}(f_{\hat{w}}) + \frac{(1+\epsilon)\sqrt{C}\sqrt{2C_2 n^{-\beta_2}}}{\sqrt{n}} + \xi \\
&\leq (\epsilon+1)\hat{L}(f_{\hat{w}}) + C_3 n^{-\beta_2} + O(n^{-\frac{3}{4}})
\end{aligned}
\tag{43}
$$

This final result can also extend to $l$-layer transformer as follows:

$$
\begin{aligned}
L(f_{\hat{w}}) &\leq (\epsilon+1)\hat{L}(f_{\hat{w}}) + \frac{(1+\epsilon)\sqrt{C}\sqrt{l\sum_{i=1}^{l}h_i}}{\sqrt{n}} + \xi \\
&\leq (\epsilon+1)\hat{L}(f_{\hat{w}}) + \frac{(1+\epsilon)\sqrt{C}\sqrt{lC_2 n^{-\beta_2}}}{\sqrt{n}} + \xi \\
&\leq (\epsilon+1)\hat{L}(f_{\hat{w}}) + C_3 n^{-\beta_2} + O(n^{-\frac{3}{4}})
\end{aligned}
\tag{44}
$$

Thus, we could finally derive the upper bound of the test loss scaling behavior as follows:

$$
L(f_{\hat{w}}) \leq (\epsilon+1)\hat{L}(f_{\hat{w}}) + C_3 n^{-\beta_2} + O(n^{-\frac{3}{4}})
\tag{45}
$$

## C. Additional Experimental Results

All fine-tuning was performed by using PyTorch and the Hugging Face Transformers library.

### C.1. Sensitivity to Hyperparamters

To further evaluate the robustness of our fine-tuning process, we conducted a series of experiments analyzing the sensitivity to key hyperparameters, including learning rates, batch sizes, and average input sequence length. In particular, we examined how variations in learning rate and batch size influence the Pearson correlation across three benchmark datasets. The results of these experiments are summarized below.

*Table 8.* Impact of Learning Rate and Batch Sizes on Pearson Correlation on FLAN

| Batch size \ Learning rate | $3e^{-5}$ | $1e^{-4}$ | $3e^{-4}$ | $1e^{-3}$ |
|---|---|---|---|---|
| 64 | 78.36 | 78.41 | 78.40 | 78.39 |
| 128 | 78.32 | 78.34 | 78.43 | 78.36 |
| 256 | 78.37 | 78.36 | 78.36 | 78.34 |

*Table 9.* Impact of Learning Rate and Batch Sizes on Pearson Correlation on Gigaword

| Batch size \ Learning rate | $3e^{-5}$ | $1e^{-4}$ | $3e^{-4}$ | $1e^{-3}$ |
|---|---|---|---|---|
| 64 | 85.74 | 85.73 | 85.74 | 85.75 |
| 128 | 85.74 | 85.79 | 85.77 | 85.76 |
| 256 | 85.69 | 85.72 | 85.71 | 85.69 |

The results demonstrate that our method exhibits strong robustness with respect to variations in learning rates and batch sizes. For the FLAN dataset, Pearson correlation remains highly stable, fluctuating only slightly within the range of 78.32 to 78.43. Similarly, the Gigaword dataset shows consistent correlation values between 85.69 and 85.79 across all configurations. Although the Wikitext dataset exhibits marginally more variability, the correlations still remain tightly clustered between 82.50 and 82.61. These findings indicate that the model's performance is largely insensitive to changes in these hyperparameters, reinforcing the reliability of our fine-tuning process.

*Table 10.* Impact of Learning Rate and Batch Sizes on Pearson Correlation on Wikitext

| Batch size \ Learning rate | $3e^{-5}$ | $1e^{-4}$ | $3e^{-4}$ | $1e^{-3}$ |
|:---:|:---:|:---:|:---:|:---:|
| 64 | 82.60 | 82.61 | 82.60 | 82.60 |
| 128 | 82.54 | 82.53 | 82.55 | 82.55 |
| 256 | 82.51 | 82.51 | 82.51 | 82.50 |

## C.2. Sensitivity to Input Length

To assess the impact of average input sequence length, we adjusted the input distribution by removing either the shortest or longest sequences, resulting in average lengths of 18 and 22 tokens, respectively, compared to the original average of 20. As shown in Table 11, both shorter and longer averages lead to slight decreases in Pearson correlation and relative accuracy. Specifically, the Pearson correlation drops from 78.14 (at average length 20) to 77.39 and 76.89 for lengths 18 and 22, respectively, while relative accuracy similarly declines. Although performance is affected by these changes, the overall variation remains small, suggesting that the model is relatively robust to moderate fluctuations in input sequence length.

*Table 11.* Impact of Average Input Sequence Length on FLAN

| Metric \ Avg. Input Seq. Length | 18 | 20 | 22 |
|:---:|:---:|:---:|:---:|
| Pearson Correlation (PearCorr) | 77.39 | 78.14 | 76.89 |
| Relative Accuracy (RelAcc) | 87.86 | 88.88 | 87.91 |

