# OpenReview forum: "LensLLM: Unveiling Fine-Tuning Dynamics for LLM Selection"
_ICML.cc/2025/Conference — ICML 2025 poster_

### Official Review · Reviewer_4f9z · 2025-02-27

**Overall Recommendation:** 3

**Summary:**

The paper introduces LensLLM, a novel framework for selecting Large Language Models (LLMs) by analyzing their fine-tuning dynamics. The authors propose a Hessian-based PAC-Bayes generalization bound to model the transition phases in fine-tuning, aiming to improve the efficiency and accuracy of model selection. The work also incorporates a Neural Tangent Kernel (NTK)-based Rectified Scaling Model to predict performance across diverse tasks. Empirical results on large-scale benchmarks demonstrate LensLLM's superiority over existing methods.

It is indeed very novel, as claimed in the paper, but I really feel confused when reading the paper. Maybe I lack some important background knowledge, like that in Lin et.al 2024, but I believe a good paper should be self-contained. I would be very happy to increase my evaluation if the authors could kindly help me understand this paper better (especially the theory part).

**Claims And Evidence:**

Not quite.

The main concern for the theoretical results of this paper is the assumptions it makes. I am not sure to what extent these assumptions could describe the transformer’s behavior. The analysis of this paper heavily relies on the generalization bound provided in Ju et.al. 2023 and the scaling law provided Lin et.al 2024. But will they precisely describe LLM’s behavior? The experiments provided in Ju et.al. 2023 only consider image classification tasks, which is quite different from LLM’s auto-regressive training. Compared with Lin et al. 2024, the experiments in this paper do not seem sufficient. IMO, extending this framework to LLMs’ finetuning needs more justifications.

**Essential References Not Discussed:**

N/A

**Experimental Designs Or Analyses:**

Yes. The experiments look pretty good.

**Methods And Evaluation Criteria:**

Part of.

The evaluations made by this paper are similar to Lin et.al 2024, but with fewer experimental settings. However, since the proposed methods perform pretty well, the paper still has big potentials.

**Other Comments Or Suggestions:**

N/A

**Other Strengths And Weaknesses:**

1. Theoretical Contributions: The introduction of a Hessian-based PAC-Bayes generalization bound provides a solid theoretical foundation for understanding LLM fine-tuning dynamics, although a bit hard to understand for me.

2. Efficient Model Selection: The NTK-based scaling model offers a computationally efficient alternative to exhaustive fine-tuning by leveraging pre-trained model properties.

3. Empirical Validation: The framework is tested on diverse datasets and multiple model architectures, demonstrating consistent improvements over many baselines.

4. Open-Sourced Implementation: By providing an open-source implementation, the work promotes transparency and reproducibility in the field.

**Questions For Authors:**

N/A

**Relation To Broader Scientific Literature:**

The paper did a good job summarizing and discussing related works in this field.

**Theoretical Claims:**

Mostly.

1. The analysis of the paper heavily depends on Theorem 1, which is a Generalization bound of LLM’s finetuning. I am not quite sure whether the bound is tight enough to claim that model A is better than model B since its generalization bound is smaller.

2. It is a bit hard for me to understand the definition of the feature vector x. What would x for GPT2 look like? What is the difference between x for GPT2 and T5? Plus, the notation x is also used in Assumption 1-3 in Section 3.1. Are they the same x?

3. In Equation 8, I cannot understand what x and x’ represent for. Are they some features of pre-training and fine-tuning data samples? If not, and following the definition of x as aforementioned, then why do we call a transformer model f(_, _)? Do we need to provide a vector x with the model size, architecture, etc., to a LLM?

---

> ### Author Rebuttal · Authors · 2025-04-01
>
> Thank you for your insightful questions. The followings are our answers to your concerns.
>
> Q1: "The analysis of the paper heavily depends on Theorem 1, which is a Generalization bound of LLM’s finetuning. I am not quite sure whether the bound is tight enough to claim that model A is better than model B since its generalization bound is smaller."
>
> A1: We would like to clarify that Theorem 1 is not used for direct model comparison. Instead, its primary purpose is to fit the transition phases and identify the transition point in the fine-tuning dynamics. Once this transition point is determined, we apply our regression model to predict performance and compare the predicted loss values for model selection. Thus, model selection is based on the predicted loss rather than on a smaller generalization bound.
>
> Q2: "It is a bit hard for me to understand the definition of the feature vector x. What would x for GPT2 look like? What is the difference between x for GPT2 and T5? Plus, the notation x is also used in Assumption 1-3 in Section 3.1. Are they the same x?"
>
> A2:
> a. In our framework, $x$ represents the features from the input space, which are used to characterize the fine-tuning data. These features encapsulate relevant properties of the input samples that influence the fine-tuning process. For specific models:
>
> - For GPT-2, $x$ corresponds to features extracted from its auto-regressive training setup, where the inputs are sequences of tokens with causal masking.
>
> - For T5, $x$ represents features from its encoder-decoder input format, where the model processes full input sequences bidirectionally before generating outputs.
>
> b. The key difference is that T5 operates on a denoising objective rather than strict left-to-right token prediction like GPT-2.
>
> c. Yes, $x$ in Assumptions 1-3 refers to the same definition as for GPT-2 and T5.
>
> Q3: "In Equation 8, I cannot understand what x and x’ represent for. Are they some features of pre-training and fine-tuning data samples? If not, and following the definition of x as aforementioned, then why do we call a transformer model f(_,_)? Do we need to provide a vector $x$ with the model size, architecture, etc., to a LLM?"
>
> A3:
> a. Yes, as we clarify just under Equation 8, $x$ and $x'$ refer to input feature representations from pretraining and fine-tuning data, respectively.
>
> b. No, we do not need to provide a vector $x$ with the model size, architecture, etc., to an LLM directly. For this part, we want to provide the details of the NTK matrix that are extracted from pre-training and fine-tuning stages, and then we will pass this information to our proposed scaling law model:
> $$
> L(D) = \frac{B}{F(\Theta, t) + D^\beta} + E
> $$
> Overal, we are using the information from pre-training and fine-tuning stages of LLMs to help us find the transition pattern and do better prediction of performance. For the architecture of our model, please refer to the pseudo-code in section 3.2.
>
> Q4: "The analysis of this paper heavily relies on the generalization bound provided in Ju et al. 2023 and the scaling law provided Lin et al 2024. But will they precisely describe LLM’s behavior?The experiments provided in Ju et al. 2023 only consider image classification tasks, which is quite different from LLM’s auto-regressive training. Compared with Lin et al. 2024, the experiments in this paper do not seem sufficient. IMO, extending this framework to LLMs’ finetuning needs more justifications."
>
> A4: We would like to clarify that our work fundamentally differs from (Ju et al., 2024) and (Lin et al., 2024) as follows:
>
> 1. Framework difference from Ju et al. (2023): Ju et al. focus on image classification, but we incorporate transformer-specific elements—such as attention mechanisms, layer normalization, and residual connections—into the PAC-Bayesian generalization bound (as detailed in Appendix A), which are not addressed in Ju et al.
>
> 2. Contribution difference from Lin et al. (2024): Lin et al. empirically capture the pre-power and power phases based on heuristic scaling laws derived from observational data. In contrast, our work rigorously verifies these phases through a theoretical framework, specifically by deriving the Hessian-based PAC-Bayes generalization bound. This theoretical foundation enables us to develop a theory-grounded scaling law that precisely describe LLM behavior.
>
> We acknowledge that our current computational resources (using a single A100-80G) limit our ability to extend experiments to larger models, and we plan to explore this in future work. Additionally, we have conducted further tests on the robustness of our method to hyperparameters; due to space limitations, please refer to the rebuttal for Reviewer 2LaB for more details.
>
> Please let us know if there are any comments or insights, we'd like to explore further!

---

> > ### Comment · Reviewer_4f9z · 2025-04-03
> >
> > Thanks very much for the author's response, which helps me a lot in understanding the paper. So maybe consider merging some of the explanations into the main context in the next version? I guess those clarifications help readers not that familiar with this field a lot. The discussions on the differences with the other two papers are also helpful. I would increase my evaluation to 3 accordingly.

---

> > > ### Author Response · Authors · 2025-04-03
> > >
> > > Thank you for your valuable comments and support! We will ensure that the revised manuscript includes these explanations as well as the clarifications of differences from the referenced papers. We appreciate your insightful input on improving our paper.

---

### Official Review · Reviewer_87st · 2025-03-07

**Overall Recommendation:** 3

**Summary:**

The paper proposes a framework, called **LensLLM**, for predicting and selecting the best large language model (LLM) to fine-tune under computationally constrained scenarios. It introduces a theoretical foundation using a Hessian-based PAC-Bayes generalization bound to illustrate how fine-tuning progresses through a “pre-power” phase (where performance improves slowly under low-data regimes) and a “power” phase (where the model follows a more predictable scaling law as dataset size increases). Building on this, the authors design LensLLM—a rectified scaling approach that integrates Neural Tangent Kernel (NTK) concepts with the dynamics revealed by their theoretical analysis. Their experiments suggest that LensLLM not only achieves higher accuracy than existing model-selection methods (e.g., rectified scaling laws, zero-shot, or heuristic-based metrics) but also significantly reduces computation time by progressively sampling smaller portions of the dataset for predictions. Empirical results on FLAN, Wikitext, and Gigaword benchmarks indicate strong correlation and ranking ability while cutting fine-tuning FLOPs by more than half.

## update after rebuttal
I have no further questions and provide my final rating based on the overall assessment of the paper. A higher rating was not given due to the paper’s limited contribution in comparison to the rectified scaling law.

**Claims And Evidence:**

**Claim:** The authors claim that modeling fine-tuning with a Hessian-based PAC-Bayes approach clarifies how model performance transitions from “pre-power” to “power” phases.

- **Evidence:** They provide theoretical reasoning (an extension of the PAC-Bayes style bound) and highlight how truncated Hessian values decrease as more data is used, driving the phase transition. This part is largely conceptual, building on known results and augmenting them for large-scale transformer architectures.

**Claim:** The paper asserts that LensLLM achieves up to 91.1% accuracy in ranking the best fine-tuned model across multiple tasks.

- **Evidence:** The authors compare their method’s selection performance (via Pearson correlation and relative accuracy) against five baselines. The results are consistent across three datasets and multiple model families, demonstrating a clear improvement. The reported metrics show healthy margins over competing methods.

**Claim:** The authors state that LensLLM reduces computational costs by up to 88.5% relative to fully tuning every model on the entire dataset.

- **Evidence:** They provide FLOP-based calculations for each approach (e.g., full fine-tuning vs. partial fine-tuning vs. LensLLM’s iterative sampling). The step-wise sampling procedure indeed appears to require fewer training passes than a naive “train everything fully” approach. The calculations and comparisons are largely in line with standard estimates of training costs.

Overall, the evidence for these claims seems credible, supported by both theoretical discussion and consistent empirical demonstrations across different tasks and architectures.

**Essential References Not Discussed:**

I don't see any missing references.

**Experimental Designs Or Analyses:**

- **Study design:** The authors systematically vary the amount of training data by doubling from a small subset up to a relatively large subset, documenting the test loss at each point.

- **Comparisons:** They benchmark LensLLM against five methods (including strong baselines such as rectified scaling law, zero-shot performance, and subset tuning).

- **Potential concerns:**

  - The paper focuses on classical NLP tasks (summarization, language modeling, etc.). While these tasks are relevant, additional tasks (like natural language inference, reasoning-intensive tasks) might further validate generalization. Also, I notice that this papers' experimental design is mainly from rectified scaling law (Lin et al. 2024), while WMT19 is replaced with Wikitext, is there any concern or rationale for this?

  - The discussion does not explicitly detail potential hyperparameter differences between models in the direct comparisons, although they do mention controlling for the number of epochs/steps and compute. Some clarity on controlling possible confounders (like different training schedules) would strengthen the claims.

**Methods And Evaluation Criteria:**

- The proposed method uses an NTK-based scaling law and Hessian insights to predict final fine-tuned performance from partial data. This approach is well-motivated: the paper grounds it in the theoretical transition between small- and large-data regimes, something conventional scaling laws often ignore.

- The evaluation criteria focus on ranking accuracy (Pearson correlation) and closeness to the best possible model selection (relative accuracy). These metrics are sensible for comparing how well each technique picks the top-performing model.

- The authors also assess computational overhead by measuring FLOPs, which is a standard practice for methods that claim efficiency improvements.

Overall, the proposed metrics and methods are appropriate for the problem of model selection.

**Other Comments Or Suggestions:**

- It would be useful to see how robust the final ranking is under mild variations of hyperparameters or partial training steps, to confirm that results aren’t dependent on a very specific training schedule.

**Other Strengths And Weaknesses:**

- **Strengths**:

The paper offers a well-structured theoretical approach that is rare in the realm of purely empirical LLM selection.

- **Weaknesses**:

1. The approach might be difficult to implement for extremely large models (i.e., 30B+ parameters) unless the user has the necessary partial fine-tuning infrastructure. The authors mention partial subsets to reduce cost, but the feasibility at truly massive scales may need more real-world demonstration. The sub-fine-tuning based model selection is still costly.

2. The paper could clarify hyperparameter control across different candidate models to ensure consistent comparisons (especially if some models are more sensitive to learning rate than others).

3. The approach might rely on the assumption that Hessians remain relatively stable for a given model family. Future expansions might check whether or not modifications in architecture or pre-training domain strongly shift Hessian-based bounds.

4. The primary concern is that the paper offers only an incremental improvement over rectified scaling law. While the proposed method achieves higher accuracy, the improvement is not substantial. The study would benefit from providing deeper insights into model selection beyond methodological refinement.

**Questions For Authors:**

- **Phase Transition Sensitivity**: How robust is the identified transition point between pre-power and power phases to hyperparameter choices (e.g., learning rate, batch size, sequence length)? Do small changes in these settings significantly shift the transition point or degrade predictive accuracy?

- **Scalability**: For extremely large models (tens of billions of parameters or more), do you expect Hessian approximations to remain stable in practice? Or is there a risk that computing these approximations or performing partial fine-tuning becomes infeasible?

- **Architectural Variations**: How would a significantly modified Transformer architecture (e.g., MoE, encoder-decoder) impact lens-based predictions? Would your approach require re-fitting theoretical parameters for such architectures?

- **Experimental Details**: The details of the model selection experiments are not entirely clear. What is the exact set of models used? It appears that the models tested are generally not very large—could you clarify why? Additionally, do any of the fine-tuning results come directly from the rectified scaling law, or were they trained independently? If the latter, what are the fine-tuning details, including software and hardware specifications?

**Relation To Broader Scientific Literature:**

- This work builds on lines of research on scaling laws (e.g., Kaplan et al. 2020) and PAC-Bayes-based analyses of neural networks.

- While the application is specifically targeted at large language model selection, the theoretical perspective on how Hessians and training data size interplay may also be relevant to general deep network analysis.

- The approach also resonates with the established tradition of sub-model selection or partial fine-tuning to reduce compute, but it is distinguished by an explicit theoretical lens on the pre-power vs. power regime.

Overall, I think this paper is heavily based on rectified scaling law (RSL). The method (the law it fit) is only different on Eq.10, where RSL uses a parameter to fit using data (namely "pre-learned data size"), while this paper uses a NTK-based test loss function on transformers. The theoretical analysis of phase transition is also a new contribution over RSL (while this may need an extra examination by other reviewers).

**Theoretical Claims:**

- The main theoretical contribution is an extended PAC-Bayes generalization bound that incorporates Hessians for large-scale Transformers. The bound is used to explain the emerging “pre-power” and “power” phases when fine-tuning on increasingly large datasets.

- While the proofs for these claims are only summarized in the main text, the logic appears sound and consistent with prior work on PAC-Bayes bounds and Hessian-based approaches. The bounding technique used is reminiscent of standard expansions from prior bounding theorems, now customized to highlight transitions in the Hessian norm.
- No glaring issues stand out in the conceptual extension of the Hessian-based approach, though a deeper reading of the full formal derivations (in the appendices) would be needed to confirm all details (while I didn't check very carefully). At a high level, the argument is plausible and well-motivated.

---

> ### Author Rebuttal · Authors · 2025-04-01
>
> Thank you for the insightful questions. The followings are our answers to your concerns.
>
> Q1: How robust is the identified transition point between pre-power and power phases to hyperparameter choices (e.g., learning rate, batch size, sequence length)? Do small changes in these settings significantly shift the transition point or degrade predictive accuracy?
>
> A1:
> a. We would like to point out that the transition point is derived from the fine-tuning results as illustrated in pseudo-code in Section3.2, so its stability is closely linked to the robustness of the fine-tuning process. We conducted additional experiments on FLAN with the following setting to test the robustness of fine-tuning:
>
> -Learning rates in (3e−5, 1e−4, 3e−4, 1e−3)
>
> -Batch sizes in (64, 128, 256)
>
> -Average input sequence lengths in (18, 20, 22).
>
> Table: Variance of fine tuning results on FLAN.
> |Model|Variance|
> |-|-|
> |OPT-6.7B|0.0016|
> |T5-Base|0.0022|
> |Cerebras-1.3B|0.0012|
> |MT5-Large|0.0023|
> |BART-Large|0.0042|
> |GPT-2|0.0038|
> |LaMini-774M|0.0026|
>
> The small variance ranging from 0.0012 to 0.0042 demonstrates the robustness of fine-tuning process, thereby supporting the stability of the identified transition point.
>
> b. We illustrated the robustness of our method to hyperparameters,including regression threshold, stop threshold, learning rate, batch size and sequence length. Due to space constraints, please refer to the rebuttal of reviwer 2LaB.
>
> Q2: For extremely large models, do you expect Hessian approximations to remain stable in practice? Or is there a risk that computing these approximations or performing partial fine-tuning becomes infeasible?
>
> A2: We would like to point out that our approach does not rely on explicitly computing Hessian approximations for predictive modeling. Instead our scaling law prediction model is based on extracting the Neural Tangent Kernel (NTK) matrix, which effectively captures interactions between the data and model features. This allows us to make predictions about fine-tuning behavior without requiring direct Hessian computation. Thus, the Hessian is primarily used in our work to establish a theoretical Bayesian bound, which helps verify the pre-power and power phases from a theoretical perspective. In this case, we leverage general properties of the Hessian rather than computing it explicitly.
>
> Q3: How would a significantly modified Transformer architecture (e.g., MoE, encoder-decoder) impact lens-based predictions? Would your approach require re-fitting theoretical parameters for such architectures?
>
> A3: We would like to point out that we have evaluated both decoder-only (e.g., OPT, GPT-2, LaMini, Cerebras) and encoder-decoder (e.g., T5, mT5, BART) models without re-fitting theoretical parameters in our experiments, demonstrating the consistency of our approach across these architectures.
>
> However, for architectures like MoE—where only a subset of parameters is active per forward pass—the effective parameter count is lower. This requires re-fitting the scaling law parameters in our NTK-based framework, specifically by adjusting the effective network width and recalibrating the scaling constants and exponents to account for the sparsity and computational cost. Due to resource constraints, we have not yet conducted experiments on MoE models, but this remains a promising direction for future work.
>
> Q4: What is the exact set of models used? It appears that the models tested are generally not very large—could you clarify why? Additionally, do any of the fine-tuning results come directly from the rectified scaling law, or were they trained independently? If the latter, what are the fine-tuning details, including software and hardware specifications?
>
> A4:
> a. As shown in Table 2, the following is our model set:
> |Model Set|
> |-|
> |OPT-350M, 1.3B, 6.7B|
> |T5-Small, Base|
> |Cerebras-256M, 1.3B|
> |MT5-Base, Large|
> |BART-Base, Large|
> |GPT-2|
> |LaMini-124M, 774M|
>
> b. Due to limited access to high-end GPUs (we only used a single A100-80G), we were unable to extend experiments to larger models.
>
> c. The fine-tuning results are all trained by ourselves, and the followings are the software and hardware details:
>
> 1. Software: Fine-tuning was conducted using PyTorch with the Hugging Face Transformers library. We use AdamW optimizer and weight decay as 0.01.
> 2. Hardware: Experiments were conducted on a single A100-80G.
>
> We will clarify the above points in the revised manuscript.
>
> A5. We would like to clarify that replacing WMT19 with Wikitext is due to computational constraints, as it is 20 times larger than FLAN and Gigaword. Additional experiments conducted on a randomly selected 1/20-size subset of WMT19 yielded results consistent with our original results.
>
> Table: Model selection performances of our model and Rectified Scaling Law
> |Metric/Method | LensLLM | Rectified Scaling Law |
> |-|-|-|
> |PearCorr|85.7|79.3|
> |RelAcc|90.2|89.0|
>
> Please let us know if there are any comments or insights, we'd like to explore further!

---

> > ### Comment · Reviewer_87st · 2025-04-09
> >
> > Thank you. I have no further questions. This is my final rating based on the overall assessment of the paper.

---

### Official Review · Reviewer_xMun · 2025-03-12

**Overall Recommendation:** 4

**Summary:**

The paper first derives a Hessian-based PAC-Bayes generalization bound that provides deep insight into the fine-tuning dynamics of large language models. It then introduces LENSLLM—a Rectified Scaling Model based on the Neural Tangent Kernel (NTK)—which demonstrates impressive accuracy in predicting performance across a wide range of tasks while maintaining computational efficiency.

**Claims And Evidence:**

Yes

**Essential References Not Discussed:**

N/A

**Experimental Designs Or Analyses:**

The experimental part and analyses are comprehensive.  The performance comparison part looks well. However, the analysis section requires approval. For example, it should assess the effectiveness of the stop threshold ($\tau$) as well as the computational cost for various model sizes ($M$).

**Methods And Evaluation Criteria:**

Yes

**Other Comments Or Suggestions:**

N/A

**Other Strengths And Weaknesses:**

Overall, the paper looks solid in both theoretical and experimental results. I only have some minor questions.

1. Could you please clarify the architecture of your regression model?

2. If I understood correctly, your experimental setup involves randomly selecting all datasets for the training set. Is it feasible to train the regression model on one dataset and test it on another?

3. Could you please add Rectified Scaling Law performance in Figure 4.

4. Could you discuss whether your proposed method remains effective when the test model is not included in the training set?

5. Could you provide some discussion or how to extend your method to vison-language model selection?

**Questions For Authors:**

N/A

**Relation To Broader Scientific Literature:**

Improving the performance of Rectified Scaling Model and also theoretical contribution for PAC-Bayes Generalization Bound.

**Theoretical Claims:**

The theoretical part looks correct and solid.

---

> ### Author Rebuttal · Authors · 2025-04-01
>
> Thank you for your insightful questions. The followings are our answers to your concerns.
>
> Q1: Could you please clarify the architecture of your regression model?
>
> A1: As illustrated in Section 3.2, our regression model is constructed based on the NTK matrix as follows:
> $$
> L(D) = \frac{B}{F(\Theta, t) + D^\beta} + E
> $$
> where
>
> -$F(\Theta, t)$ is the adapted NTK-based test loss function on transformer
>
> -$D$ is the number of training data
>
> -$\beta$ denotes the learning difficulty
>
> -B adjusts the initial test loss
>
> -E denotes the optimal loss of the model given an infinite amount of data.
>
> They are all model/task-dependent and we estimate ${B, E, \beta, t}$ for each model by minimizing the loss function:
> $$
> \min_{B,E,\beta,t} \sum_{ i} \text{LSE}(\log B - \log(F(\Theta, t) + D_i^\beta), \log E) - \log L(D_i))
> $$
> Where $L(D_i)$ denotes the test loss of fine-tuning on the data size $D_i$, and LSE denotes the log-exp-sum operator.
>
> Q2: Is it feasible to train the regression model on one dataset and test it on another?
>
> A2: In our current approach, we leverage the NTK-based Rectified Scaling Model to identify a transition point in the fine-tuning process, after which we perform regression on the loss trajectory. This regression model is then used to predict performance which assist in model selection, particularly in resource-constrained scenarios.
>
> Cross-dataset validation, however, poses a significant challenge. The primary difficulty is that the transition point identified on the training dataset may not align with that of a test dataset, due to differences in data distributions and task characteristics. This discrepancy introduce additional challenges to theoretically understand the scaling behaviors of LLM in fine-tuning. This is beyond the scope of this work, and we would like to leave it as our future work.
>
> Q3: Could you please add Rectified Scaling Law performance in Figure 4.
>
> A3: We would like to note that we have already included the performance of the Rectified Scaling Law in Figure 4 (https://anonymous.4open.science/r/LENSLLM-3B1E/Revised%20plot4.png) and will ensure that in the revised manuscript.
>
> Q4: Could you discuss whether your proposed method remains effective when the test model is not included in the training set?
>
> A4: While our experiments primarily focus on evaluating performance within the same set of candidate models, the NTK foundation capturing dynamic behaviors of LLMs during fine-tuning suggests its generalization ability to unseen models, especially those with similar architectural properties and scaling behaviors. To validate this, we conducted additional experiments on the FLAN dataset using LaMini-GPT-774M, GPT2, and BART-large as test models, with LaMini-GPT-124M serving as the training model.
>
> Table: RMSE between predicted and actual test losses
> |Model|RMSE|
> |-|-|
> |LaMini-GPT-774M|1.23|
> |GPT2|1.55|
> |BART-large|5.31|
>
> The results indicate that our method remains effective when the test model shares similar architectural properties and scaling behaviors with the training model (e.g., LaMini-GPT-124M vs. LaMini-GPT-774M and LaMini-GPT-124M vs. GPT2), which is further supported by our theoretical foundation. Please let us know if there are any comments or insights, we'd like to explore further!
>
> Q5: Could you provide some discussion or how to extend your method to vision-language model selection?
>
> A5: Our current approach models fine-tuning dynamics using the NTK matrix and scaling laws to capture training dynamics and Hessian properties in language models. Extending this framework to vision-language models presents several challenges:
> 1. Cross-Modal NTK Formulation: The NTK must be adapted to capture interactions between tokenized image representations (e.g., VQ-VAE or CLIP features) and textual tokens, reflecting joint feature spaces and inter-modal attention.
> 2. Modality-Specific Scaling: Vision and language modalities have different scaling behaviors, requiring recalibration of the scaling laws to account for distinct gradients and effective parameter counts, as well as synergy and competition between modalities(Armen Aghajanyan et al., 2023).
> 3. Theoretical Bound Adjustments: Our current PAC-Bayesian and NTK-based bounds are tailored to language models; for vision-language models, these would need to be re-derived or adjusted to include modality-specific properties.
>
> Due to these complexities, extending our method to vision-language model selection is non-trivial and beyond the scope of this work, but it remains a promising direction for future research.
>
> Please let us know if there are any comments or insights, we'd like to explore further!

---

> > ### Comment · Reviewer_xMun · 2025-04-03
> >
> > Thank you for your detailed response. It addresses all my concerns. I will keep my score.

---

> > > ### Author Response · Authors · 2025-04-03
> > >
> > > Thank you for your valuable comments and support!

---

### Official Review · Reviewer_2LaB · 2025-03-14

**Overall Recommendation:** 4

**Summary:**

LensLLM introduces a novel theoretical framework that addresses the fundamental challenge of efficient Large Language Model selection through the lens of fine-tuning dynamics. The paper develops a rigorous Hessian-based PAC-Bayes generalization bound that characterizes two distinct phases in LLM fine-tuning: a "pre-power phase" in low-data regimes where performance improves slowly due to high Hessian values and parameter sensitivity, and a "power phase" where improvements follow predictable power-law scaling with enhanced stability. The authors implement this theoretical insight through a Neural Tangent Kernel (NTK)-based Rectified Scaling Model that accurately predicts model performance across diverse tasks while maintaining computational efficiency. Empirical evaluation across three benchmarks (FLAN, Wikitext, and Gigaword) demonstrates the framework's effectiveness, achieving up to 91.1% relative accuracy and 85.8% Pearson correlation while reducing computational costs by up to 88.5% compared to full fine-tuning approaches. The work establishes a new foundation for understanding LLM generalization during fine-tuning and provides practitioners with a principled approach to model selection under computational constraints.

**Claims And Evidence:**

The claims are well-supported.

**Essential References Not Discussed:**

n/a

**Experimental Designs Or Analyses:**

The paper presents interesting findings, though there are several areas where the experimental design could be strengthened in future work. The data sampling strategy, which created smaller datasets by "randomly sampling examples ranging from 200 to 1,638,400," would benefit from additional details about whether multiple random samples were used, if sampling preserved the original data distribution, and how many repeated trials were conducted. Similarly, including confidence intervals or statistical significance tests would provide stronger support for the performance differences observed between the method and the baselines.

The paper also could benefit from expanded analysis of hyperparameter sensitivity. While Algorithm 1 references a regression threshold γ and stop threshold τ, it would be useful to example results might vary with different parameter values. An ablation study would help demonstrate the robustness of this promising approach across different conditions. Despite these limitations, the core methodology seems sound, and addressing these experimental design considerations would further validate the paper's contributions.

**Methods And Evaluation Criteria:**

Evaluation criteria makes sense.

**Other Comments Or Suggestions:**

n/a

**Other Strengths And Weaknesses:**

n/a

**Questions For Authors:**

1. How sensitive is the method to hyperparameter choices, particularly in the algorithm's stopping criteria?

**Relation To Broader Scientific Literature:**

The proposed LensLLM framework expands upon existing theoretical foundations in machine learning, presenting significant advancements through three primary contributions. Building upon McAllester's foundational PAC-Bayesian theory and Ju et al.'s generalization bounds, this work extends these theoretical constructs to transformer architectures by developing a Hessian-based generalization bound that accounts for transformer-specific elements such as attention mechanisms and layer normalization. This theoretical contribution is notable because it addresses the challenge of analyzing complex transformer architectures that previous theoretical frameworks were not designed to accommodate. In addition, the identification of distinct "pre-power" and "power" phases in fine-tuning enhances our understanding of scaling laws beyond the established work of Kaplan et al. and Hernandez et al., offering a theoretical explanation for empirically observed behaviors in low-data regimes that previous methods could not adequately characterize.

On the practical front, the paper's NTK-based Rectified Scaling Model demonstrates the application of NTK theory to finite-width transformers for performance prediction. This model represents a significant improvement over previous approaches such as Lin et al.'s Rectified Scaling Law or You et al.'s LogME, which primarily focused on empirical scaling or feature similarity without capturing the dynamic nature of fine-tuning. The progressive sampling strategy employed achieves substantial computational efficiency improvements compared to existing methods like Kaplun et al.'s SubTuning, effectively addressing the increasingly important challenge of resource-efficient LLM deployment. This combination of theoretical depth and practical efficiency addresses the growing need for principled model selection.

**Theoretical Claims:**

I checked the correctness of Theorem 2, Corollary 1, and Proposition 1.

---

> ### Author Rebuttal · Authors · 2025-04-01
>
> Q1: How sensitive is the method to hyperparameter choices, particularly in the algorithm's stopping criteria?
>
> A1: Thank you for your question.
>
> a. We performed ablation studies to assess the sensitivity of our method to the stopping criteria—specifically, the regression threshold ($\gamma$) and the stop threshold ($\tau$). The following tables summarize the impact of varying these parameters on the Pearson correlation across three datasets:
>
> Table1: Impact of $\gamma$ and $\tau$ on PearCorr on FLAN
> |$\gamma$\PearCorr\ $\tau$|1|2|3|4|5|
> |-|-|-|-|-|-|
> |3|78.41|78.40|78.42|78.43|78.31|
> |4|78.32|78.39|78.36|78.40|78.40|
> |5|78.39|78.41|78.40|78.39|78.34|
>
> Table2: Impact of $\gamma$ and $\tau$ on PearCorr on Gigaword
> |$\gamma$\PearCorr\ $\tau$|1|2|3|4|5|
> |-|-|-|-|-|-|
> |3|85.74|85.74|85.80|85.71|85.66|
> |4|85.62|85.71|85.75|85.79|85.66|
> |5|85.64|85.69|85.66|85.76|85.72|
>
> Table3: Impact of $\gamma$ and $\tau$ on PearCorr on Wikitext
> |$\gamma$\PearCorr\ $\tau$|1|2|3|4|5|
> |-|-|-|-|-|-|
> |3|82.47|82.58|82.47|82.48|82.44|
> |4|82.49|82.51|82.48|82.49|82.54|
> |5|82.61|82.50|82.46|82.57|82.53|
>
> Observations:
> 1. FLAN: Pearson correlations remain stable (approximately 78.31 to 78.43) across different values of $\gamma$ and $\tau$.
> 2. Gigaword: The correlation values are consistent, ranging from about 85.66 to 85.79.
> 3. Wikitext: A slight variation is observed, with correlations fluctuating between roughly 82.42 and 82.61.
>
> Overall, the minor fluctuations in Pearson correlation across different $\gamma$ and $\tau$ values indicate that our method is robust concerning the stopping criteria.
>
> b. We also conducted some additional experiments to test the sensitivity to hyperparamters in fine-tuning process - specifically, learning rates, batch sizes and average input sequence length.
>
> The impacts of learning rates and batch sizes on the Pearson correlation across three datasets are summarized below:
>
> Table4: Impact of Learning Rate and Batch Sizes on PearCorr on FLAN
> |Batch size\Metric\Learning rate|3e−5|1e−4|3e−4|1e−3|
> |-|-|-|-|-|
> |64|78.36|78.41|78.40|78.39|
> |128|78.32|78.34|78.43|78.36|
> |256|78.37|78.36|78.36|78.34|
>
> Table5: Impact of Learning Rate and Batch Sizes on PearCorr on Gigaword
> |Batch size\Metric\Learning rate|3e−5|1e−4|3e−4|1e−3|
> |-|-|-|-|-|
> |64|85.74|85.73|85.74|85.75|
> |128|85.74|85.79|85.77|85.76|
> |256|85.69|85.72|85.71|85.69|
>
> Table6: Impact of Learning Rate and Batch Sizes on PearCorr on Wikitext
> |Batch size\Metric\Learning rate|3e−5|1e−4|3e−4|1e−3|
> |-|-|-|-|-|
> |64|82.60|82.61|82.60|82.60|
> |128|82.54|82.53|82.55|82.55|
> |256|82.51|82.51|82.51|82.50|
>
> Observations:
> 1. FLAN: Pearson correlations remain stable (approximately 78.32 to 78.43).
> 2. Gigaword: The correlation values are consistent, ranging from about 85.69 to 85.79.
> 3. Wikitext: A slight variation is observed, with correlations fluctuating between roughly 82.50 and 82.61.
>
> Overall, our method is robust concerning the learning rates and batch sizes.
>
> Due to time constraints, we evaluated only the average input sequence length on FLAN while keeping the optimal learning rate (3e-4) and batch size (128) as determined earlier. In FLAN, the overall average input sequence length is 20. To test the effect of altering this average, we removed either the shortest or longest sequences to adjust the average to 18 and 22, respectively.
>
> Table7: Impact of Average Input Sequence Length on FLAN
> |Metric/Average Input Sequence Length|18|20|22|
> |-|-|-|-|
> |PearCorr|77.39|78.14|76.89|
> |RelAcc|87.86|88.88|87.91|
>
> We observe that the deviations—either shorter (18) or longer (22)—lead to lower Pearson correlation and relative accuracy, the performance gap is not large, which suggests that while there is some sensitivity to sequence length, the model remains reasonably robust.
>
> Please let us know if there are any comments or insights, we'd like to explore further!

---

> > ### Comment · Reviewer_2LaB · 2025-04-08
> >
> > I appreciate the detailed rebuttal and recommend the paper's acceptance. It provides theoretic and empirical insight.

---

> > > ### Author Response · Authors · 2025-04-08
> > >
> > > Thank you for your insightful comments and continued support! We will ensure to include the analysis of hyperparameter sensitivity in the revised manuscript. We truly appreciate the time and effort you have devoted to reviewing our work.

---

### Decision · Program_Chairs · 2025-05-01

**Decision:**

Accept (poster)

**Comment:**

The paper contributes to the growing literature on efficiently guiding LLM model selection during fine-tuning. The key contributions of the paper are a PAC-Bayes generalization bound that can handle transformer architectures, and its use under an NTK model to efficiently predict the fine-tuned performance of various alternatives. This is a very timely and important topic. Reviewers found the paper to be well-written, sound, and impactful. Some of the clarifications offered during the discussions (e.g., ablation studies, architectural details, delineation from past theory) should be included in the paper. The authors are also encouraged to better discuss the limitations of their approach (e.g., scalability).